# Artificial intelligence literacy, sustainability of digital learning and practice achievement: A study of vocational college students

**Xuefei Lin**[ID][1,2], **Guangyu Xu**[ID][3]*, **Bin Xiong**[1,2¤]

**1** School of Mathematical Sciences, East China Normal University, Shanghai, China, **2** Shanghai Key Laboratory of Pure Mathematics and Mathematical Practice, Shanghai, China, **3** School of Mathematical Sciences, Chongqing Normal University, Chongqing, China

¤ Current address: East China Normal University, No. 500 Dongchuan Road, Minhang District, Shanghai, China
* 52205500036@stu.ecnu.edu.cn

## Abstract

The rapid expansion of AI and the massive use of digital learning are creating a huge change in higher education. In contrast to general higher education, which is at the center of change, the changes in vocational higher education do not seem to have received sufficient attention from researchers. Focusing on artificial intelligence literacy and sustainability of digital learning competences, this study examined the operational behaviors of 1004 students in a practical course in a higher education institution in mainland China by using the COM-B theory. The results showed that AI literacy was positively correlated with sustainable digital learning ability, and AI literacy was positively correlated with students' sustainable digital learning behaviors through sustainability of digital learning competences. However, students in vocational institutions are not able to translate this learning behavior into a path to achieve good performance in practice, and even sustainability of digital learning competences can be slightly counterproductive. There is no excessive effect on the model after controlling for variables such as gender, family resources, and ethnic minorities, which contributes to benefit equally from quality education.

## Introduction

The pursuit of UN Sustainable Development Goal 4 (Quality Education) advocates for inclusive and equitable quality education for all [1] but there are gaps in its reality, gaps that both developed and developing countries. For instance, the proportion of the world's youth who are not in education, employment or training (NEET, Nyet) has increased from 21.8% in 2015–2019 to 23.3% in 2020, according to the UN Sustainable Development Goals Report 2022 [2]. This makes it necessary to focus on the education and training among young people.

**Data availability statement:** All relevant data are within the paper and its Supporting information files.

**Funding:** This research was supported by Shanghai Key Laboratory of Pure Mathematics and Mathematical Practice. (Fund No.: 22DZ2229014).

**Competing interests:** The authors have declared that no competing interests exist.

To bridge these educational gaps and improve accessibility and effectiveness, innovation in educational tools and approaches is essential. One of the most significant recent developments in this regard is the emergence of Generative Artificial Intelligence (GAI) [3–5].

In parallel with the advancement of GAI, the concept of Artificial Intelligence literacy has gained increasing attention as a critical foundation for navigating this new digital landscape. Artificial Intelligence literacy has become a core competency in modern education, encompassing not only the technical understanding of AI systems but also critical thinking, ethical reflection, and collaborative interaction. UNESCO [1] and Ng [6] propose a widely adopted framework that defines AI literacy through four dimensions: understanding technology and its impact, usage and collaboration, evaluation and self-reflection, and ethical awareness. While existing research has explored AI literacy in K-12 and higher education settings—emphasizing conceptual acquisition, critical evaluation, and integration into teaching practices [6–8]—its application in vocational education remains limited. Studies have also extended the concept to include public engagement and lifelong learning, highlighting the importance of making AI literacy accessible to broader populations [9,10]. Given the lack of AI-related prerequisites in vocational education, assessing students' AI literacy requires careful consideration of their ability to understand, apply, reflect on, and ethically evaluate AI technologies. Building on Ieva [8] and UNESCO's guidelines [11], this study adopts a multidimensional framework that balances the foundational content of K-12 programs [12–14] with the self-reflective and ethical depth emphasized in higher education [15–18].

Building upon this foundation, the notion of sustainable digital learning competence has emerged as a complementary construct that extends beyond AI-specific knowledge to encompass broader digital and behavioral capabilities essential for long-term learning success in technology-rich environments [19]. Sustainable digital learning competence is increasingly regarded as a critical construct in digital-era education, integrating digital competence, learning competence, and sustainable behavioural patterns [20]. As defined by ICILS and endorsed by organisations like UNESCO and OECD [21,22], digital competence encompasses the ability to use digital tools creatively and effectively across various life domains, while learning competence involves motivation, knowledge, and application [23,24]. When sustained over time, these abilities contribute to what psychologists consider habitual behaviours [25], forming the basis of sustainable learning practices. Building on this foundation, the current study adopts and adapts established measurement frameworks from Ličen [26] and Yang [27], operationalizing sustainable digital learning competence across four dimensions: digital literacy, digital skills, digital interaction, and technology integration. These dimensions reflect the evolving expectations of modern learning environments, where meaningful digital engagement and the integration of technological tools are essential to fostering student participation, collaboration, and long-term learning outcomes [28–30].

In the field of education, the rise of Generative Artificial Intelligence (GAI) has triggered a dramatic wave of digital transformation. This emerging technology offers

the potential to provide transformative solutions, such as enabling teachers and students to customize learning materials more efficiently. However, it also introduces significant challenges, particularly ethical concerns and risks of exacerbating educational inequities [15,31–36]. For instance, GAI may further disadvantage students in under-resourced areas where access to ICT infrastructure remains limited.

Despite increasing attention to AI and digital transformation in both basic and higher education, student groups in higher vocational education—especially those who are non-mainstream or marginalized—continue to receive inadequate coverage in related research, policy discourse, and institutional support [37]. Compared to comprehensive universities, vocational institutions in China remain relatively disadvantaged in terms of digital infrastructure, teaching capacity, and curricular resources [38]. Additionally, most existing research focuses on general education or STEM fields [39], with limited exploration of how vocational students interact with AI tools and digital learning environments [40].

This neglect is compounded by the longstanding marginalization of vocational education in China. Culturally and institutionally, vocational tracks are often stigmatized as inferior pathways, associated with low social status and limited career prospects. Studies by Jiang et al. [41] and Chen [42] have highlighted the widespread parental and societal reluctance toward vocational routes. Furthermore, the absence of standardized curricula in vocational education [43] means that digital competencies, including AI literacy, are often insufficiently developed. As a result, vocational students are positioned as a disadvantaged group within the educational system.

Such educational marginalization makes vocational students a key focus group for promoting the United Nations Sustainable Development Goals (SDGs), particularly SDG 4 (Quality Education), SDG 5 (Gender Equality), SDG 9 (Industry, Innovation, and Infrastructure), and SDG 10 (Reduced Inequalities) [44]. Educational equity under the SDG framework emphasizes not only access, but also the real opportunity for all learners—regardless of family socioeconomic status, gender, or ethnicity—to benefit from quality education [45,46]. Yet, international large-scale assessments such as PISA consistently reveal persistent disparities in educational outcomes linked to these background factors [46–49].

Moreover, there remains a critical gap between students' use of personal digital technologies and their understanding of how to engage with ICT tools for sustainable learning purposes [50,51]. Without the guidance and support to use AI appropriately, these technologies risk deepening existing inequalities rather than bridging them.

Taken together, these issues point to a pressing need to investigate the intersection of AI literacy, digital learning behavior, and educational equity in vocational settings. This study, therefore, aims to examine whether—and how—these structural inequalities extend into students' digital learning behaviors and achievements, thereby contributing to a more inclusive and equitable educational landscape.

To address these shortcomings and promote educational equity, this paper focuses on a marginalized student group in China's higher vocational education and explores the following issues:

RQ1: Is there a correlation between AI literacy and sustainable digital learning competences?

RQ2: Can artificial intelligence literacy and sustainable digital learning competences influence students' learning behavior? Does this behavior facilitate or help students translate it into practical skills?

RQ3: Can artificial intelligence literacy and sustainable digital learning competences empower all learners (regardless of family socioeconomic status, gender, ethnicity, etc.) to benefit equally from quality education?

## Literature review and hypotheses development

### AI literacy

The concept of AI literacy has been defined by a large number of scholars, and according to UNESCO [1], Ng [6], it can be defined in four parts: technology and impact, use and collaboration, evaluation and self-reflection, and ethics and morality.

Based on Ng's [6] description of awareness and impact, the 27 article conceptualises AI literacy in the following form: educating learners to master the concepts, skills, knowledge and attitudes underlying knowledge, but without any a priori

knowledge base. In addition to being the end user of an AI application, learners should also understand the technology behind it. Burgsteiner et al. [52] and Kandlhofer et al. [53] define AI literacy as the ability to understand the basic technologies and concepts behind AI in different products and services. In addition, some researchers have linked AI literacy to perceptual ability, confidence and readiness to learn AI. However, since there are only relatively few studies applying AI literacy to vocational education approaches, we can only integrate and explore this concept from K-12 education, vocational education and higher education for a more accurate explanation, for example, in the k-12 section, Druga et al. [7] and Lin et al. [54] design learning lessons and activities to develop AI literacy, focusing on how learners acquire AI concepts. This section of the higher education perspective seems to go even deeper, requiring that teachers and students must understand the fundamentals of AI, such as machine learning at the foundational level, in order to critically assess when and how to incorporate AI tools into pedagogy, student assessment and evaluation, or to better facilitate the delivery of courses and learning [6,8,16].

AI enhances human intelligence through digital automation, and 19 articles [6] mention that AI literacy can enable learners to participate in higher-order thinking activities and generate better interactions. In addition to understanding and using the concepts and practices of artificial intelligence, some studies have also extended AI literacy to two other abilities, enabling individuals to critically evaluate AI technology and effectively communicate and collaborate with AI [9]. For example, Han et al. [55] improved students' knowledge of science and technology, and then applied it to scientific research-based learning to solve practical problems. Long et al. [10] encourage citizens to participate in the co creation of AI facilities in public places to broaden their public AI literacy and experience. Participants can participate in the production of public interactive art works, from being initially attracted by AI devices to interacting with the devices and others. Overall, although these articles have slightly different definitions of AI literacy, they all support the view that everyone, especially K-12 children, has acquired basic AI knowledge and abilities, enhanced motivation for future careers, and utilized AI technology [56].

In higher education, this interactivity is more evident, as AI technology is widely applied in the interaction between teachers and students, such as teaching and completing specific tasks [8,16–18]. In vocational education, based on the sparse existing literature, Ferhat [57] et al. combined the nursing profession with AI to integrate academic support mechanisms into the digital transformation process of educational programmes.The work done by Ferhat appears to be in a state between the simple situational interactions of K-12 and the specific complex interactions of higher education, whereby the teacher performs the appropriate pedagogical interventions in anticipation of the student completing the specific and paradigmatically fixed operational processes.

For the assessment and self-reflection phase it seems that K-12 appears to be somewhat inadequate. Many people claim to know little about AI [9]. Nevertheless, "folk theories" (i.e., "informal theories...... sense and explain how systems work") to explain AI algorithms [58]. These theories, whether accurate or not, shape the nature of user interaction and experience [58] which seems to be reduced to a previous stage of use and collaboration. A better understanding of how AI works can help people form more accurate mental models of the systems they interact with. Higher education has shown more interest in this area, and Dong [9] conducted a review of topics covered in university syllabi, including machine learning [12,13], AI [14] writing up a curriculum A list of all topics covered in the curriculum was written, as well as a list of learning objectives for the K-12 audience in the AI education program [7,59]. Topics range from high-level concepts (e.g., learning, kinematics, planning) to concrete implementations (e.g., Bayesian networks, Markov models). Most syllabi are aimed at computer science majors, and many courses require that K-12 programs also require students to have some prerequisite knowledge of mathematics, statistics, or computer science. This level of prerequisite knowledge may make these courses and their content inaccessible to groups that could benefit from AI literacy, such as children who interact with AI at home or adults who use AI in the workplace [60]. Considering the specificity of higher vocational institutions and the lack of pre-learning for students about AI courses, it is important for assessment and self-reflection to focus more on assessing the wider societal impact of AI, including its potential benefits and risks, and how it may affect education and future work.

Ethical Adoption UNESCO's [11] first global guidance on GenAI in education aims to support countries in taking immediate action, planning long-term policies, and developing human capacity to ensure human-centered guidance on these new technologies. The guidance assesses the potential risks that GenAI may pose to the promotion of core human values of human agency, inclusiveness, equity, gender equality, linguistic and cultural diversity, and plurality of views and expressions. It proposes key steps for government agencies to regulate the use of GenAI tools, including mandating data privacy protections and considering age limits for use. It outlines requirements for GenAI providers to ensure their ethical and effective use in education. The guide emphasizes the need for educational institutions to validate the ethical and pedagogical applicability of GenAI systems in education. It calls on the international community to reflect on its long-term impact on knowledge, teaching, learning, and assessment. The publication provides concrete recommendations for policymakers and educational institutions on how to design the use of GenAI tools to protect human subjectivity and truly benefit learners, teachers, and researchers.

In summary, the questionnaire used for the AI literacy of students in higher vocational education addresses the four dimensions of influence, cooperation, self-reflection, and ethics, adapted from Ieva [8], UNESCO [11], etc., and balances the basic content from the K-12 [9,12–14]and the self-reflection from the perspective of higher education [8,16–18].

## Sustainability of digital learning competences

Sustainability digital learning competences need to be defined in an aggregated manner in three dimensions, namely digital competence, learning competencies and sustainability behaviours. Digital Competence (DC) has been interpreted in a number of ways [61], with the International Institute for Computing and Information Literacy (ICILS) definition of digital competence as an individual's ability to use computers for investigative, creative, and communicative purposes in order to participate effectively at home, school, workplace, and in the community being commonly used. Learning competence is the way in which individuals (and organisations as groups of individuals) identify, assimilate and use knowledge [62]. It is widely recognised in the academic community that learning competence consists of three elements: goals (motivation), will and ability (knowledge and practice) [63]. Aggregating the above two definitions yields a definition of digital learning competence repeatedly called for by several organisations such as UNESCO and OECD [22], defining digital learning competence as a set of knowledge, skills and attitudes that enable students to use digital tools effectively for learning in digital learning environments [27]. Sustainable behaviours are behaviours that a person adopts over a long period of time or periodically, which are often referred to as 'habits' by psychologists [25]. A student is considered to have sustainable behaviour if he/she is able to adopt a certain behaviour for more than 21 consecutive days.

Based on the above basic concepts and the definitions of digital learning competencies provided by organizations such as UNESCO and the OECD (i.e., a set of knowledge, skills, and attitudes that enable students to effectively use digital tools for learning in digital learning environments), sustainable digital learning competences (SDLC) can be defined as: the ability of individuals to develop habitual skills for long-term or regular use of digital resources and frequent and effective application of digital tools for learning in digital learning environments, based on certain digital competencies and learning abilities.

Based on the above definitions, we combined Ličen [26] and Yang [25] scales for determining digital learning ability to adapt them to finalize our measurement of students. Sustainability digital learning competence is finally measured by four dimensions: data literacy, data skills, digital interaction, and technology integration which are composed of four dimensions.

Digital literacy is a critical and evolving competency that goes beyond the basic use of digital tools. For example, proficiency in the use of digital tools, integrating digital simulations, and facilitating interactions between students through digital platforms are becoming increasingly important in modern education [26,28]. The ongoing digital transformation of higher education further emphasizes the importance of digital literacy as an essential element of effective learning. Digital literacy is dynamic and must evolve as new technologies emerge [29,64].

Digital skills are another important dimension that focuses on the technical skills required to effectively use various digital tools and platforms in teaching. These skills include using learning management systems, digital collaboration tools, and specialized software that supports online and blended learning environments. Teachers and students must not only be proficient in using digital tools to enhance student engagement and learning outcomes [26]. This will create a wealth of learning opportunities for both teachers and students, which aligns with a broader digital competency framework that emphasizes the importance of specific digital skills for effective teaching and learning [29,65].

Digital interaction focuses on the use of digital tools to facilitate interactive learning environments. Technology integration refers to the incorporation of digital tools and resources into instructional practices to enhance learning. These two dimensions are critical for creating meaningful connections between students, teachers and content. The ability to interact and collaborate using digital platforms is increasingly recognized as the foundation of digital competence, especially in higher education where student engagement is a key factor in academic success [30,66,67]. This is in line with a growing body of research that assesses the role of digital tools in creating dynamic learning environments that promote active learning and collaboration [68].

## Specificity of vocational institutions and educational equality

Vocational institutions are not part of the mainstream values in China. According to Jiang et al.'s [41] study of the student population in Chinese vocational institutions, students have a very low status among the public and parents do not want their children to pursue specialized studies such as vocational technology. Chen [42] is even more outspoken in stating that vocational education is a dead end in the eyes of Chinese parents. This shows that this group is at a disadvantage in Chinese culture. In addition, vocational education lacks curriculum standards [43], and it seems that AI literacy and digital learning skills simply do not get any representation here., this group has almost become a disadvantaged group, so it is beneficial to conduct research here to promote the United Nations Sustainable Development Goals (SDGs), especially SDG 4 (Quality Education), SDG 5 (Gender Equality), SDG 9 (Industry, Innovation, and Infrastructure), and SDG 10 (Reducing Inequality) [69].

Specifically, educational equity in SDG is seen as equality of ability, or a real opportunity to realize the function of education [45,46], but this does not seem to be well achieved. According to the PISA [46] survey with previous studies, student achievement varies across family socioeconomic status (SES), gender, and across ethnicity [47–49]. For example, family socioeconomic status (SES) had different levels of influence on different subjects, with r = 0.202 for math, while verbal achievement accounted for r = 0.271. gender differences [70] gave females the upper hand in verbal achievement, but the academic ability gap was not significant. In contrast, differences in the performance of ethnic minority student groups have received attention in the last century, and this group tends to have a more disadvantaged socioeconomic status and distinct cultural characteristics, and it has been demonstrated that ethnic minority students are at a disadvantage in terms of performance compared to majority students [71].

In conclusion, in order to promote equity (SDG 5, SDG 10), it is hoped that factors such as family socioeconomic status, gender, and ethnic minorities do not have an impact on students' digital learning behaviors and practical achievements.

## COM-B theory framework

There are a number of models to consider in the study of sustainable digital learning capabilities and students' sustainable learning behaviours. For example, the Technology Acceptance Model (TAM) Unified theory of acceptance and use of technology (UTAUT) have both been applied to the study of students' intentions towards AI. These models have been widely acted on undergraduate student population by researchers. For example, Le Dang [72] in analysing the intentions of university students to use AI for learning, Yu [73] in studying the behavioural intentions of university students to use next-generation information technology, including AI, in intelligent foreign language learning, and Chatterjee and

Bhattacharjee [74] in investigating the effects of behavioural intentions on AI adoption in higher education. It seems that both models have very relevant capabilities to undertake this task, but all of them share a rather interesting point: it is not possible to analyse all independent variables by integrating them into a single capability or a single literacy in a comprehensive consideration and behaviour. Therefore, we wanted to find a model that could address this issue at a larger or more macro level. Not only that, but these two models can only study behavioural intentions, not behaviour itself, and for this reason the COM-B model was introduced.

COM-B aims to provide an overarching model that encompasses all the factors known to influence behaviour change [75]. The model was proposed by the UK's National Institute for Health and Care Excellence (NIHCE) in 2011, citing the Capabilities, Opportunities, Motivations, Behaviours (COM-B) model [75] as a key theoretical framework for understanding and supporting behaviour change [76].The COM-B model incorporates six elements that are believed to drive behavior, namely physical ability (having skills, strength and endurance); mental ability (being able to engage in necessary thought processes such as comprehension and reasoning); physical opportunity (provided by the environment, including time and resources); social opportunity (provided by interpersonal influences, social cues, and the way we think about things, such as the words and concepts that make up language); reflexive motivation (the ability to make a conscious intention, plan and making assessments); and automatic motivation (emotional reactions, impulses, and desires) [75,77]. This model is more macroscopic and allows for the three COMs to be examined separately as latent variables or merged together to collectively support as COM, which is very much in line with the indicator of digital learning competence for sustainability as this competence is an aggregate term as per the previous discussion, which prefers a holistic conceptualization of COM merged together [78].

### Hypotheses

**The relationship between AI literacy and sustainability of digital learning competences.** Before the emergence of artificial intelligence literacy, the term 'digital literacy' was used to evaluate basic concepts and skills related to computers, which began to be popular in various industries in the 1970s. Users must be proficient in using computer systems related to their specific tasks or work. As more and more people rely on computer technology to develop new social and economic opportunities, the importance of digital literacy is increasing. This concept gradually evolved into the digital literacy that follows [79].It follows that early digital literacy is a prerequisite for AI.. But as digital transformation becomes more and more successful, the concept of digital competence has been introduced, and new ways of connecting data, people and processes in modern digital conditions aim to create better environments and prepare for future challenges, in which case AI is one of the factors influencing the creation of a digital education infrastructure.

For example, the use of AI in vocational education can be observed in the practice of engineering students, who are able to predict, monitor, optimise and plan the construction design of civil engineering works, which is similar in other engineering fields [80], where the students have mastered AI in the course of their professional training [81], and in the use of which can be verified the progress of the students [82]. Not only that, but Gonzalez [83] also mentions that each professor should be allowed to use ICT [84,85], which allows for a more effective presentation of the assessment [86,87] of each student's work. In summary, the hypotheca is made:

H1: AI literacy is beneficial for the improvement of sustainability of digital learning competences.

**The relationship between SDLC and behavior.** COM needs to be integrated with the previous SDLC, and based on the definition of COM, as well as the question set in the SDLC questionnaire and the previous discussion, it is argued here that data literacy is the competence, data skills are the opportunity, and digital interaction, and technology integration are the motivations.

The COM-B model represents the observation that at any given moment, a particular behavior will only occur if the person involved has the ability and opportunity to engage in that behavior and is more motivated to perform it than any other behavior. Sustainable digital learning behaviors are considered here as endemic behaviors. In order to measure this

behavior, Ieva's [8] formulation of the behavior is used and combined with some of the statements made by Romero [88] in his study of AI usage behavior.

According to the COM-B model, we can make the following hypothesis:

H2: SDLC promotes students' digital learning behavior.

But given the externalized nature of the behavior, and in order to explore the validity of the behavior and make up for the lack of AI literacy and digital competence measurements, while, it seems that the use of AI and the use of digitization can make some tangible results, some form of externalization is still missing [10,55,56]. we're going to make a bolder guess:

H3: Students' digital learning behavior may promote their practical achievement.

Summarizing the previous arguments leads to the final hypothesis:

H4: AI literacy promotes students' digital learning behavior by enhancing the sustainability of digital learning competences.

H5: AI literacy promotes students' practical achievement by enhancing the sustainability of digital learning competences.

## Conceptual framework

Based on the discussion in the previous section, the artificial intelligence literacy questionnaire developed by UNESCO Ieva [8] will be used to measure artificial intelligence literacy. This questionnaire consists of four sections: impact, collaboration, self-reflection, and ethics. The sustainability of digital learning capabilities (SDLC) will be measured using the digital learning capabilities questionnaire developed by Li Cen [26] and Yang [27]. Ieva's [8] and Romero's [88] will be used to determine students' digital learning behaviors, and practice scores will be used to identify the externalized forms of these behaviors. SDLC behaviors are studied based on the COM-B theoretical framework, resulting in the final conceptual framework shown in Fig 1 below

## Methodology

### Ethical considerations

This study has obtained ethical approval from the Human Subject Research Ethics Committee of East China Normal University (Approval No. HR695–2024). All data is treated with confidentiality. All participants who are willing to participate in this study have obtained informed consent. According to the requirements of the ethics review committee, this article belongs to Class II exemption and does not involve sensitive information of the subjects. As the questionnaire is conducted online, an informed consent button is set in the questionnaire title. If you choose not to agree, the questionnaire will automatically end.

### Sample

A total of 1004 first-year students (male = 869; female = 134) from a higher vocational and technical school in Liaoning Province participated in this study, with a mean age of 19.93 years (standard deviation = 2.645). To ensure the objectivity of the study, the researchers will not be directly involved in day-to-day teaching, but will only make suggestions on curriculum design for AI use and digital learning for frontline teachers to enquire about. Only this vocational institution can be used for this study as there are not many schools in the region that have instructional designs for courses on AI use and digital learning. The authors, with the help of local teachers, conducted the study with a completely randomly generated sample of all students from four colleges in all 13 colleges that offer courses in AI and digital learning. In order to protect the privacy of the students, anonymity was used to fill in the data collection and complete freedom was guaranteed to the students who would not be affected in any way if they did not wish to participate in the questionnaire completion. The

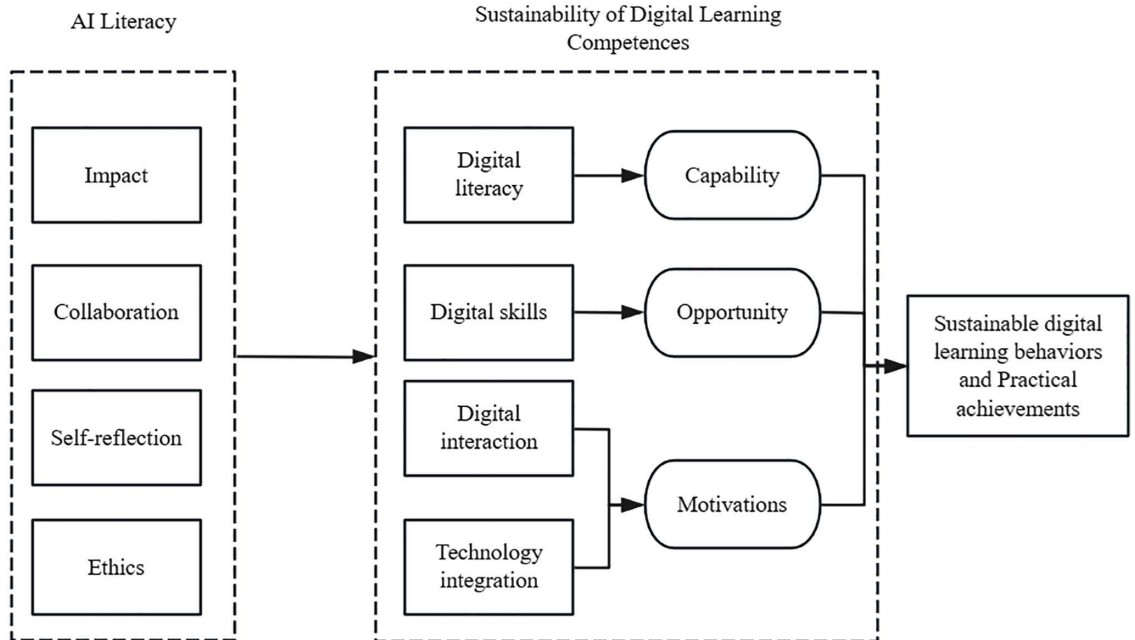

**Fig 1. Conceptual framework.**

students were predominantly Han, Manchu, Mongolian and Korean. The case school is a very prestigious vocational and technical school in Liaoning Province, whose teaching quality and student academic performance are among the highest in the region in the same field, and whose student employment rate is close to 100 per cent. The average socio-economic status of the students' families is at the average level of Liaoning Province. The proportion of families with at least one person with a master's degree or higher is 10.6 per cent, the proportion of families with the highest university or bachelor's degree is 21.7 per cent, the proportion of families with the highest high school or equivalent degree is 30.9 per cent, and the proportion of families with the highest junior high school degree is 36.9 per cent. It is particularly noteworthy that on the item of educational attainment, the sample is basically consistent with the actual results of the Seventh National Population Census of the People's Republic of China, and the sample is highly representative [89]. All the specifics are shown in Table 1 below.

## Questionnaire

This study used a questionnaire to investigate students' AI literacy and digital learning skills, and a five-point Likert scale (from "completely disagree" to "completely agree") to assess students' level of agreement with these items. The questionnaire was ethically vetted to ensure that no personal privacy was involved. The questionnaire was designed through Questionnaire Star (https://www.wjx.cn/), which consisted of thirty-three questions, of which the ten questions in the first section were basic information about the individual, which contained some basic variables that needed to be controlled. Students' gender (male = 1; female = 2), age, ethnicity, parents' highest level of education (middle school = 1; high school or middle vocational school = 2; high vocational school(associate degree) or bachelor = 3; master's degree and above = 4), household resources and book ownership (i.e., how many books in the home), and number of books in the home) were included in the analyses as covariates, with book ownership, due to the relative prevalence of modern paperlessness, being the most common for the number of e-books was also included in the analysis as the number of books to control for its effect on motivation and achievement. Family resources can represent the family socioeconomic status;

**Table 1. Sample Description (N = 1004).**

| Category | categories | Frequency | Percentage |
|---|---|---|---|
| Gender | Male | 869 | 86.6% |
| | Female | 134 | 13.4% |
| Age | 16 | 121 | 12.1% |
| | 17 | 121 | 12.1% |
| | 18 | 112 | 11.2% |
| | 19 | 113 | 11.3% |
| | 20 | 106 | 10.6% |
| | 21 | 109 | 10.9% |
| | 22 | 90 | 9.1% |
| | 23 | 105 | 10.5% |
| | 24 | 126 | 12.5% |
| Parents' highest education level | middle school | 370 | 36.9% |
| | high school or middle vocational school | 310 | 30.9% |
| | high vocational school (associate degree) or bachelor | 218 | 21.7% |
| | master or higher | 105 | 10.6% |
| Minorities or Han | Han | 678 | 67.6% |
| | Man | 105 | 10.5% |
| | Mengguo | 105 | 10.5% |
| | Chaoxian | 115 | 11.5% |
| Family book collection | less than 10 | 204 | 20.8% |
| | 10-50 | 361 | 36.2% |
| | 50-100 | 178 | 16.7% |
| | over 100 | 261 | 26.3% |

book ownership was utilized for the measurement, not only that, but personal cell phones were not considered here, this is because filling out the questionnaire was disseminated using cell phone WeChat groups, which were filled out using the students' personal cell phones.

**Artificial intelligence literacy.** Artificial intelligence literacy was the second part of the questionnaire. This questionnaire was adapted from assessment of AI literacy [8,9,11,16,17]. The teacher section was removed and only the student section was studied. Due to the large number of questions, we filtered the questionnaire by selecting only the three questions with the highest reliability for each dimension. The questionnaire was divided into four dimensions as follows, Impact, Ethics, Collaboration and Self-reflection, the details of which are shown in Table 2 below. Not only that, but these four dimensions measure the dimension of AI literacy very well.

The validation factor analysis was designed to test the validity of the four-factor structure of AI literacy values, $\chi^2(df)=94.670(48)$, $p < .001$, $\chi^2/df = 1.972 < 3$, $RMSEA = .031$, $GFI = .985$, $RFI = .983$, $CFI = .994$, $IFI = .994$, $TLI = .992$ and indicate an acceptable model fit. Its structural validity is shown in Table 3 below.

**Sustainability of digital learning competences (SDLC).** Sustainable digital learning competences (SDLC) were the third part of the questionnaire. The questionnaire was adapted from Ličen and Prosen's assessment of teachers' competencies for sustainable digital learning [26] and Yang's [27] digital learning competencies questionnaire for students in 2024. The questionnaire is similar to the questionnaire for AI literacy, which was assessed for teachers, so adaptations were made to the teacher section to make it suitable for student research. Due to the large number of questions, we

**Table 2. Reliability Analysis of AI Literacy Scale.**

| Variables | Cronbach's alpha | Items |
|---|---|---|
| Impact (IM) | 0.754 | 3 |
| Ethics (ET) | 0.824 | 3 |
| Collaboration (CO) | 0.840 | 3 |
| Self-reflection (SR) | 0.742 | 3 |
| **AI Literacy** | **0.940** | **12** |

**Table 3. Validity Testing of Differences in Various Dimensions of AI Literacy Scale.**

| Variables | IM | ET | CO | SR |
|---|---|---|---|---|
| IM | **0.528** | | | |
| ET | 0.391 | **0.609** | | |
| CO | 0.397 | 0.669 | **0.636** | |
| SR | 0.407 | 0.68 | 0.684 | **0.511** |
| Sqrt of AVE | **0.727** | **0.78** | **0.797** | **0.715** |

filtered the questionnaire and selected only the three questions with the highest reliability on each dimension. The questionnaire was categorized into the following four dimensions; Digital Literacy, Digital Skills, Digital Interaction, and Technology Integration, which are shown in Table 4. Not only that, but these four dimensions also measure well the dimension of sustainable digital learning competences.

The validation factor analysis was designed to test the validity of the four-factor structure of digital learning values, $\chi^2(df) = 57.770(48)$, $p < .001$, $\chi^2/df = 1.204 < 3$, $RMSEA = .014$, $GFI = .990$, $RFI = .991$, $CFI = .999$, $IFI = .999$, $TLI = .998$ and indicate an acceptable model fit. Its structural validity is shown in Table 5 below.

**Sustainable digital learning behavior and practice achievement.** Prior to the distribution of the questionnaire during the spring semester of 2024, students underwent specialized training led by three instructors, focusing on the application of artificial intelligence and digital tools in learning. Concurrently, practical hands-on courses were also implemented in parallel. The training program comprised four courses, with core content covering key modules such as digital resource retrieval and the application of artificial intelligence technology. Students were explicitly required to use artificial intelligence or digital tools for learning activities each week after class, conducting independent searches for relevant materials to support and guide the practical operation courses.

The teaching cycle for courses related to artificial intelligence and digital tools spanned the entire semester. After systematic training, students have developed the ability to independently utilize artificial intelligence and digital resources. Based on the 12-week training principle [25], it can be determined that they have developed sustainable digital learning capabilities.

**Table 4. Reliability Analysis of Digital Learning Scale.**

| | Cronbach's alpha | Items |
|---|---|---|
| Digital Literacy (DL) | 0.819 | 3 |
| Digital Skills (DS) | 0.759 | 3 |
| Digital Interaction (DI) | 0.832 | 3 |
| Technology Integration (TI) | 0.865 | 3 |
| **Digital Learning** | **0.949** | **12** |

**Table 5. Validity Testing of Differences in Various Dimensions of Digital Learning Scale.**

| Variables | DL | DS | DI | TI |
|---|---|---|---|---|
| DL | **0.602** | | | |
| DS | 0.678 | **0.529** | | |
| DI | 0.691 | 0.724 | **0.622** | |
| TI | 0.688 | 0.713 | 0.712 | **0.684** |
| Sqrt of AVE | **0.776** | **0.727** | **0.789** | 0.827 |

Following the completion of the aforementioned training and practical courses, the research team organized a final assessment for the practical courses. To ensure the objectivity and professionalism of the assessment results, the test scores were evaluated by three expert instructors with associate senior professional titles, using a percentage-based scoring system (with 60 points as the passing threshold). Since the talent cultivated by this institution primarily targets the heavy industry sector, the relevant practical operations involve numerous high-risk procedures. Improper execution could pose risks to personal safety, leading to stricter evaluation criteria—if experts determine that an operation poses a safety hazard, the test is immediately deemed failed, resulting in a relatively high overall failure rate for the course.

The specific scoring calculation method is shown in Equation (1) below:

$$x_{score} = \frac{1}{3} \sum_{i=1}^{3} x_i \quad 0 \leq x_i \leq 100$$

(1)

$x_i$ is an independent rating for each teacher. Among them, 323 students failed this test (32.17%), and the remaining 681 passed the test (67.83%), and among the groups that passed the test, the group with a score of 60–79 was 327 (32.57%), and the group with a score of 80–100 was 354 (35.26%); it is very interesting to note that each of the three groups of failing and good and good each accounted for more than one-third. A reasonable explanation for this is that since there are three teachers grading, if two of them give a low score, they will get a direct fail; if two of them give a pass and one of them gives a fail, they will get a good score; and if all three of them give a pass, they will get a high score.

After the practical assessment was completed, the above questionnaire was distributed, and students self-assessed their sustainable digital learning behavior in the questionnaire. This dimension consisted of three questions adapted from Ieva and Daniela [8] to measure the use of artificial intelligence and digital behavior. Their specific information is as follows:

1. I am aware that there are different AI tools and digital resources that can be used in different educational courses (e.g., ChatGPT, Tiangong, Doubao, Learning Power, MOOC).

2. I have tried at least once a week to use AI tools and digital resources to plan or participate in the learning process to support my learning process (e.g., preparing lesson ideas, creating presentations, fact-checking).

3. I use AI tools and digital resources to support the learning process and find a variety of AI tools and a variety of digital resources based on different learning needs and abilities, with a basic understanding of their potential mechanisms (e.g., providing text-to-speech tools, using AI tools to help check answers, finding learning courses that can support manipulation, etc.).

These three topics appeared to have a Cronbach coefficient that was not very high, and in order to understand the deeper reasons why this arose, we did a short, small-scale informal interview, and a plausible explanation is that many students were aware of the possibility of using these techniques and applications to facilitate their practical learning, but

it was difficult to find these practical exercises on the web, which made the topics less reliable. After discussion between several researchers, we unanimously decided to still use these data, this is because the content of this questionnaire has been developed by other researchers [8] and has achieved relatively good reliability and there is literature to support this claim [90,91].

## Data analysis

The questionnaires were pre-set to exclude those that were randomly filled out, and those that took less than 100 seconds to answer, or that had the same value for all questions, were simply deleted. In addition to this, no missing values will occur as each question has a mandatory answer requirement.

First, we utilized SPSS to perform descriptive statistics for the questionnaire. Second, we used Structural Equation Modelling (SEM) and maximum likelihood estimation of AMOS to examine the relationship between AI Literacy, Digital Learning and practice achievement. In this study, $\chi^2/df < 3$, $RMSEA < .08$ $GFI > .90$, $RFI > .90$, $CFI > .90$, $TLI > .90$ [78] indicated an acceptable model fit. Third, we will utilize AMOS to investigate path salience and some further exploration.

## Results

The conclusions of this paper are divided into four parts; the first part examines the SEM model between AI literacy and digital learning; the second part examines the SEM model of AI and digital learning and student behavior; the third part examines the model of AI literacy and student achievement in practice; and the fourth part concludes the multi-cluster analysis.

### Structural equation model 1

In Model 1, two variables, AI literacy and sustainable digital learning competences, were included in the study. The SEM results showed that our hypothesized model fitted the data well: $\chi^2(df) = 375.106(251)$, $p < .001$, $\chi^2/df = 1.494 < 3$, $RMSEA = .022$, $GFI = .969$, $RFI = .978$, $CFI = .993$, $IFI = .993$, $TLI = .993$.

Our results predict a very strong positive correlation between AI literacy and sustainable digital learning competences ($\beta = .998$, $p < .001$). In the section on AI literacy, each observable variable has a relatively good explanation of AI literacy. For the impact section, each of the three topics explained AI literacy positively ($\beta_1 = .554$, $p < .001$, $\beta_2 = .786$, $p < .001$, $\beta_3 = .806$, $p < .001$). For the ethics section, each of the three topics was positively linked to an explanation of AI literacy ($\beta_1 = .785$, $p < .001$, $\beta_2 = .789$, $p < .001$, $\beta_3 = .779$, $p < .001$). For the collaboration section, each of the three topics was positively linked to an explanation of AI literacy ($\beta_1 = .780$, $p < .001$, $\beta_2 = .786$, $p < .001$, $\beta_3 = .822$, $p < .001$). For the self-reflection section, each of the three topics was positively associated with an explanation of AI literacy ($\beta_1 = .795$, $p < .001$, $\beta_2 = .510$, $p < .001$, $\beta_3 = .788$, $p < .001$).

In the section Sustainable digital learning competences, each observable variable can be reasonably explained. For the digital literacy component, each of the observed variables was positively associated with SDLC ($\beta_1 = .781$, $p < .001$, $\beta_2 = .779$, $p < .001$, $\beta_3 = .777$, $p < .001$). For the digital skills section, each of the three topics was positively linked to an explanation of SDLC ($\beta_1 = .800$, $p < .001$, $\beta_2 = .543$, $p < .001$, $\beta_3 = .798$, $p < .001$). For the digital interaction section, each of the three topics was positively associated with an explanation of SDLC ($\beta_1 = .800$, $p < .001$, $\beta_2 = .787$, $p < .001$, $\beta_3 = .782$, $p < .001$). For the technology integration section, each of the three topics was positively linked to an explanation of SDLC ($\beta_1 = .787$, $p < .001$, $\beta_2 = .889$, $p < .001$, $\beta_3 = .801$, $p < .001$) Fig 2.

### Structural equation model 2

In Model 2, AI literacy and sustainable digital learning competencies and student behaviors were included in the study. The SEM results showed that our hypothesized model fitted the data well: $\chi^2(df) = 529.050(322)$, $p < .001$, $\chi^2/df = 1.643 < 3$, $RMSEA = .025$, $GFI = .961$, $RFI = .971$, $CFI = .990$, $IFI = .990$, $TLI = .989$.

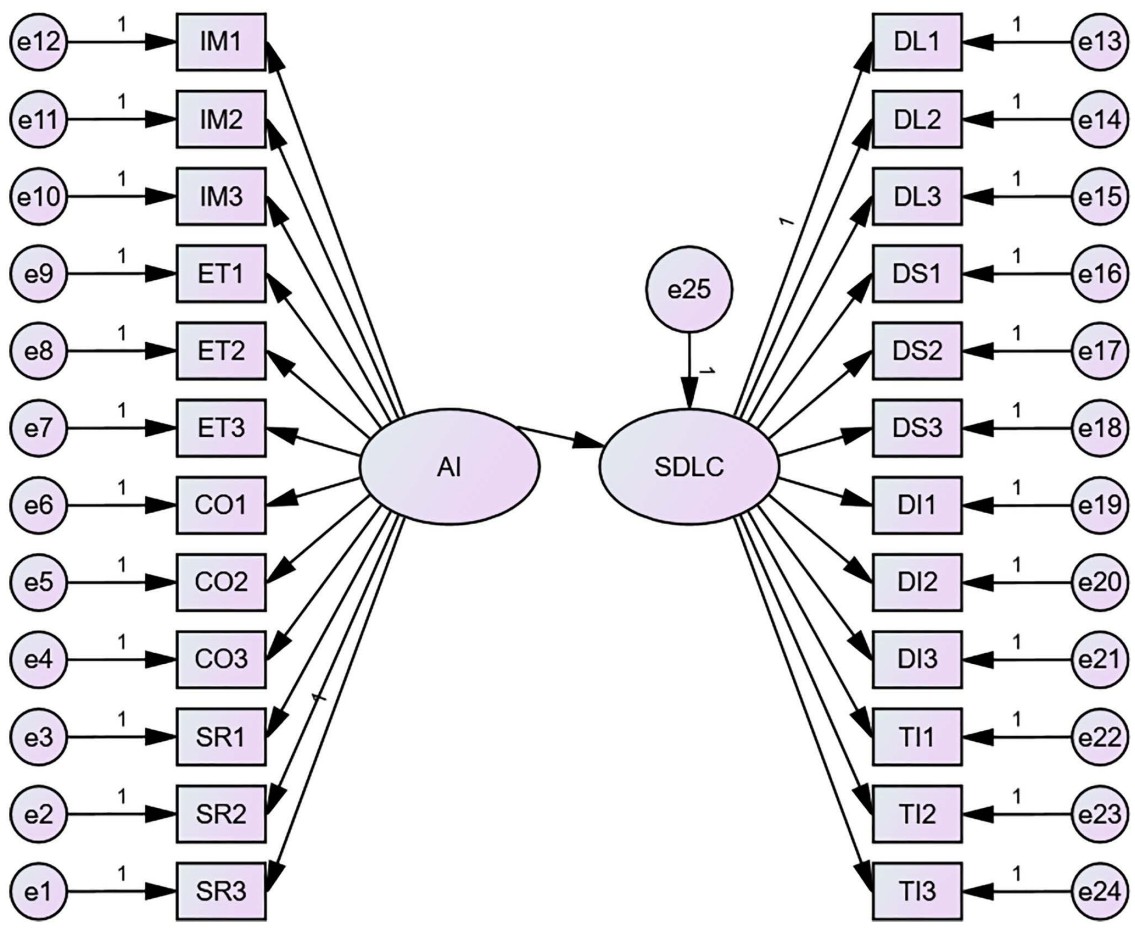

**Fig 2. SEM of AI Literacy and Sustainable Digital Learning Competences.**

In this model, the positive correlation between AI literacy and sustainable digital learning competencies was further corroborated ($\beta = .999, p < .001$), and achievable digital learning competencies also produced a significant positive correlation with student behavior ($\beta = .825, p < .001$).

In the section on AI literacy, each observable variable has a relatively good explanation of AI literacy. For the impact section, each of the three topics explained AI literacy positively ($\beta_1 = .558, p < .001, \beta_2 = .786, p < .001, \beta_3 = .806, p < .001$). For the ethics section, each of the three topics was positively linked to an explanation of AI literacy ($\beta_1 = .785, p < .001, \beta_2 = .789, p < .001, \beta_3 = .778, p < .001$). For the collaboration section, each of the three topics was positively linked to an explanation of AI literacy ($\beta_1 = .779, p < .001, \beta_2 = .785, p < .001, \beta_3 = .822, p < .001$). For the self-reflection section, each of the three topics was positively associated with an explanation of AI literacy ($\beta_1 = .795, p < .001, \beta_2 = .514, p < .001, \beta_3 = .788, p < .001$). In the section Sustainable digital learning competences, each observable variable can be reasonably explained. For the digital literacy component, each of the observed variables was positively associated with SDLC ($\beta_1 = .780, p < .001, \beta_2 = .780, p < .001, \beta_3 = .776, p < .001$). For the digital skills section, each of the three topics was positively linked to an explanation of SDLC ($\beta_1 = .798, p < .001, \beta_2 = .547, p < .001, \beta_3 = .797, p < .001$). For the digital interaction section, each of the three topics was positively associated with an explanation of SDLC ($\beta_1 = .800, p < .001, \beta_2 = .787, p < .001, \beta_3 = .783, p < .001$). For the technology integration section, each of the three topics was positively

linked to an explanation of SDLC ($\beta_1 = .787, p < .001, \beta_2 = .888, p < .001, \beta_3 = .801, p < .001$). It can be seen that in contrast to Model 1, the standardized estimates of each observable variable are more stable and reflect the latent variables well. For student behavior, the three observables provide good positive feedback ($\beta_1 = .633, p < .001, \beta_2 = .614, p < .001, \beta_3 = .675, p < .001$).

The significant paths are shown by Fig 3 below.

**Structural equation model 3**

In Model 3, AI literacy and sustainable digital learning competencies and practice achievement were included in the study. The SEM results showed that our hypothesized model fitted the data well: $\chi^2(df) = 395.781(273), p < .001,$

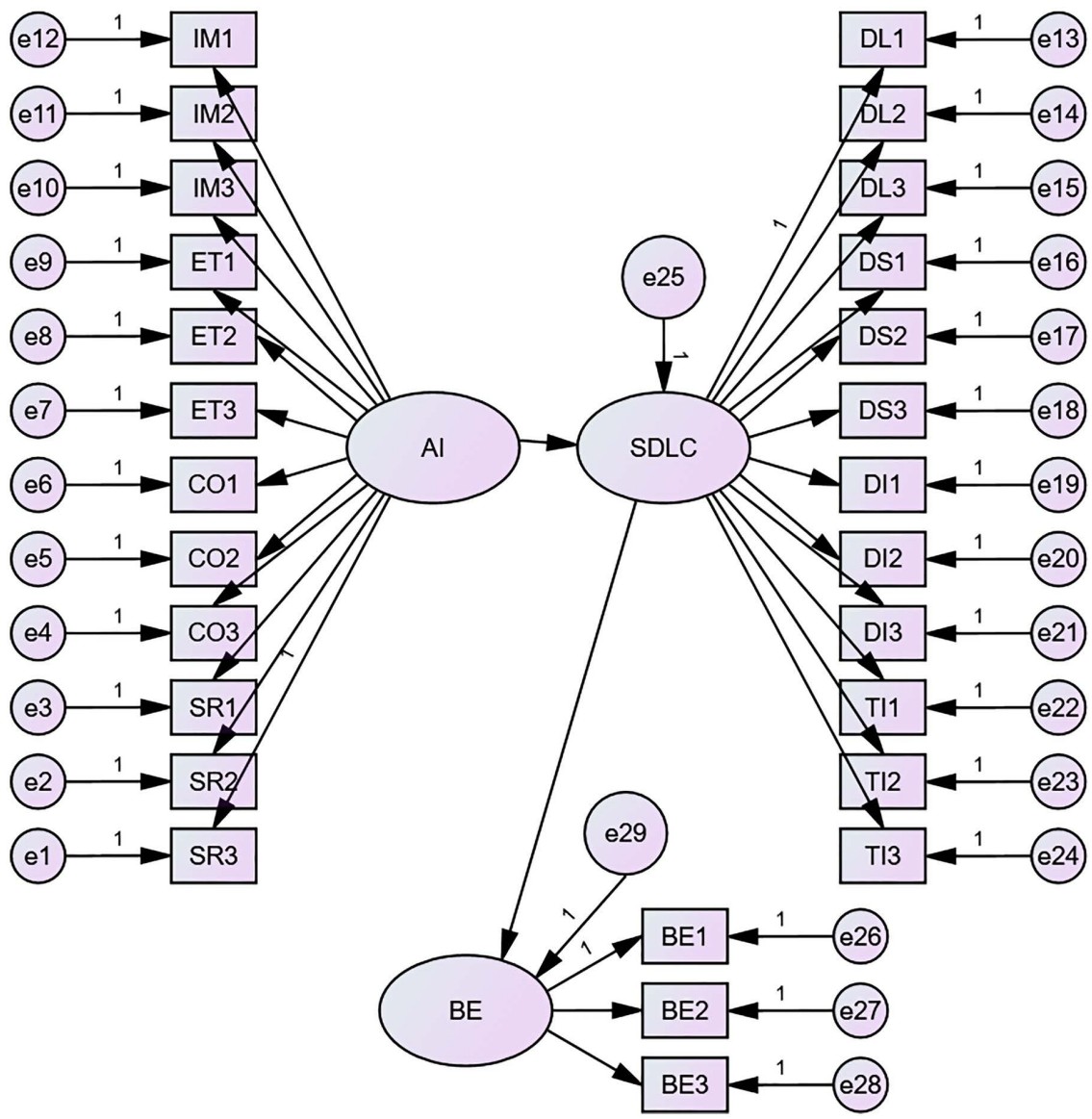

**Fig 3. SEM of AI Literacy, Sustainable Digital Learning Competencies and Behavior.**

$\chi^2/df = 1.450 < 3$, $RMSEA = .021$, $GFI = .969$, $RFI = .977$, $CFI = .993$, $IFI = .993$, $TLI = .993$. In this model, there is a positive correlation between AI literacy and sustainable digital learning competencies ($\beta = .998, p < .001$), and a positive correlation with practice achievement ($\beta = .390, p = .904$). Digital learning competencies showed a negative correlation with practice grades ($\beta = -.394, p = .903$). Unfortunately, however, the paths to practice achievement were not significant for either AI literacy or SDLC. It can be assumed that student selves have been using AI-type technology tools to design and learn much of the content, but from a practical standpoint, such help or feedback may have had minimal effect Fig 4.

## Control variables analysis

A multi-cluster analysis was performed for Model 2.

**Gender.** For gender, the model is stable. The fit is good for unconstrained models: $\chi^2(df) = 915.774(644)$, $p < .001$, $\chi^2/df = 1.422 < 3$, $RMSEA = .021$, $RFI = .951$, $CFI = .986$, $IFI = .986$, $TLI = .985$. Not only that, the unconstrained model passes the hypothesis test (measurement weights $\chi^2(df) = 23.798(24)$, $p = .473$). A similar situation occurs with measurement weights ($\chi^2(df) = 939.572(688)$, $p < .001$, $\chi^2/df = 1.407 < 3$, $RMSEA = .020$, $RFI = .952$, $CFI = .986$,

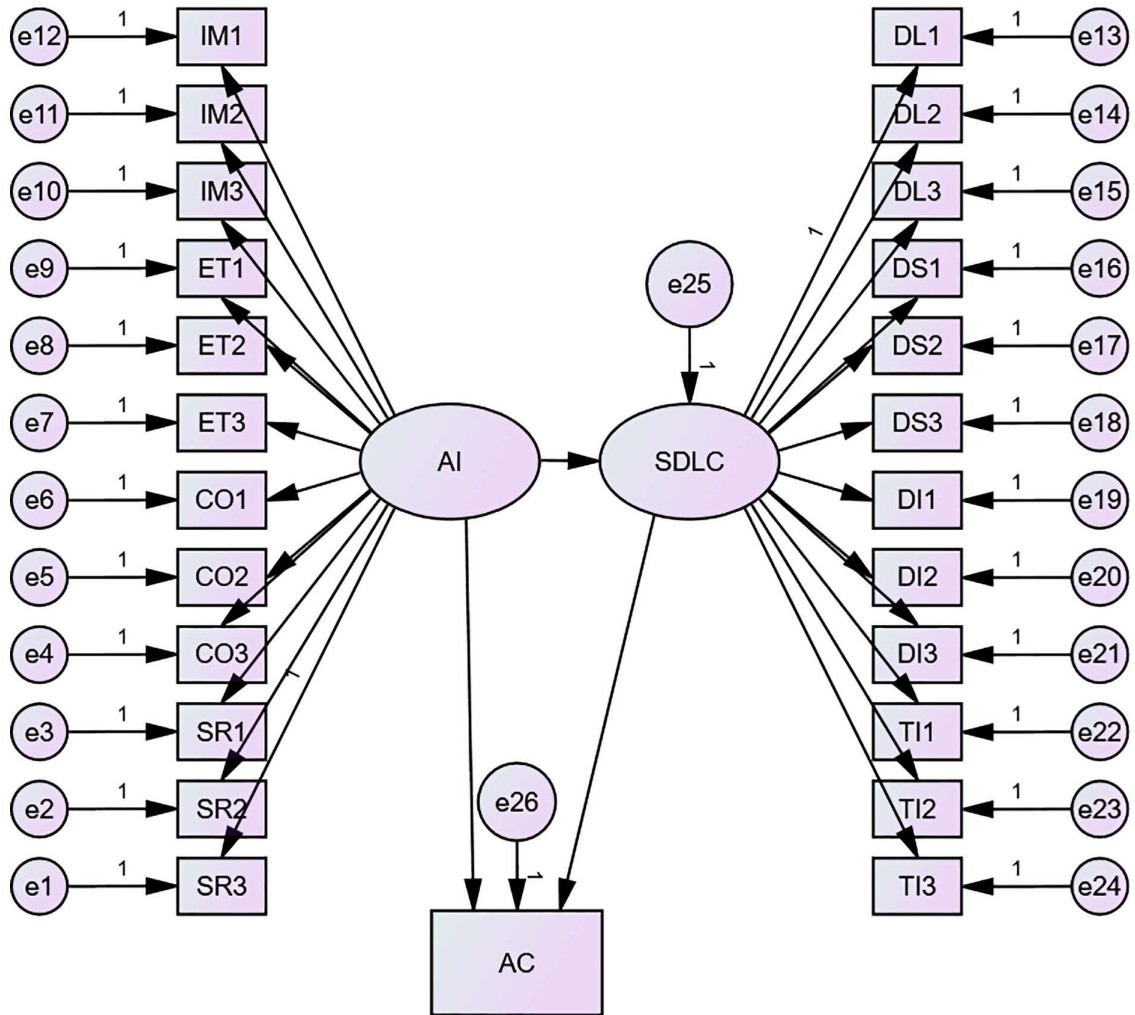

**Fig 4. SEM of AI Literacy, SDLC and Practice Achievement.**

$IFI = .986$, $TLI = .986$) and with structural weights ($\chi^2(df) = 941.168(670)$, $p < .001$, $\chi^2/df = 1.404 < 3$, $RMSEA = .020$, $RFI = .952$, $CFI = .986$, $IFI = .986$, $TLI = .986$).

**Parents' highest education level.** The highest level of parental education also appeared to be insensitive to the model created. The fit is good for unconstrained models: $\chi^2(df) = 1722.556(1400)$, $p < .001$, $\chi^2/df = 1.230 < 3$, $RMSEA = .015$, $RFI = .918$, $CFI = .984$, $IFI = .984$, $TLI = .984$. Not only that, the unconstrained model passes the hypothesis test (measurement weights $\chi^2(df) = 18.462(24)$, $p = .780$). A similar situation occurs with measurement weights ($\chi^2(df) = 1741.018(1424)$, $p < .001$, $\chi^2/df = 1.223 < 3$, $RMSEA = .015$, $RFI = .919$, $CFI = .984$, $IFI = .984$, $TLI = .984$) and with structural weights ($\chi^2(df) = 1724.178(1426)$, $p < .001$, $\chi^2/df = 1.222 < 3$, $RMSEA = .015$, $RFI = .919$, $CFI = .984$, $IFI = .984$, $TLI = .984$).

**Minorities or Han.** Minority or Han Chinese had no effect on the unconstrained model, but had a slight effect on the path. The fit is good for unconstrained models: $\chi^2(df) = 1843.115(1400)$, $p < .001$, $\chi^2/df = 1.317 < 3$, $RMSEA = .018$, $RFI = .914$, $CFI = .978$, $IFI = .978$, $TLI = .978$. The biggest difference comes from the Manchus, whose path is somewhat different from the other three ethnic groups. For example, the estimates on SDLC to student behavior ($\beta = .819$, $p < .001$) are slightly lower than the other three ethnic groups ($\beta = .828$, $p < .001$).

It can be seen that the model 2 is not sensitive to several clusters, which is a greater indication of the inclusiveness and fairness of AI literacy and Sustainable digital learning competences.

## Discussion

Overall, both the AI literacy questionnaire and the SDLC questionnaire had good reliability and validity, and the four sub-dimensions were able to clearly measure the whole question. Several structural equation models fit well and explained the previous hypotheses completely.

### AI literacy and sustainable digital learning competencies

The results showed a strong positive correlation between AI literacy and sustainability digital learning ability, an observation that is consistent with previous research [10]. Furthermore, this result has been widely and accurately demonstrated in Models 1, 2, and 3. For example, similar to the previous use of AI in vocational education for engineering students [80], students have already mastered AI [81] during their professional training and can consciously use AI to improve their digital learning abilities. However, from an impact perspective, this concept is highly volatile for the measurement of AI literacy because the questions are more conceptually oriented, an issue that cannot be avoided in either K-12 [7,54], or higher education contexts [6,8,16]. Although we preferred a K-12 setting in this section that focuses on learners acquiring the concept of AI, considering that the level of the students is not sufficient to support the condition of having a high level of math and basic prior knowledge, in terms of impact despite the greater success, there is still room for upward mobility that can enable students to look at the concept of AI in terms of discarding a part of their foundation in math. This coincides with some previous research on how to design questions to critically assess when and how to incorporate AI tools into pedagogy, student assessment and evaluation, or to better facilitate the delivery of courses and learning [6,8,16].

A similar situation to the AI impact aspect occurs in the self-reflection section, where large swings occur. This is more related to the K-12 section for this issue, for example Long [10] is outspoken about how little is known about this section, some folk theories to establish algorithms for AI which, whether accurate or not, shape the nature of user interactions and experiences [58], which seems to be reduced to the previous stage of use and collaboration. This issue has been transformed in the context of higher education, but given the level of students, this level of prerequisite knowledge may make these courses and their content inaccessible to groups that may benefit from AI literacy [60], so the question is being designed to think more about the broader societal implications of AI, including its potential benefits and risks, and how it may affect education and the future work, which brings the question back to the K-12 level when targeting students in higher vocational institutions, ultimately leading to some volatility.

This situation does not seem to exist in terms of ethics and morals, probably thanks to the fact that the content used is adapted from UNESCO [11], which makes the title free of such problems due to the large sample size and the authority of the experts. Collaboration also has a very good explanation for AI literacy, which is highly similar to many previous studies [9,10,55]. As Chai [56] states: everyone, especially K-12 children, have gained basic AI knowledge and competence, enhanced motivation for future careers, and utilized AI technology.

## Sustainable digital learning competencies and behavior

In Model 2, SDLC has a very strong and positive effect on students' digital learning behavior. Not only that, there is a further increase in the positive correlation between AI literacy and SDLC. This is consistent with the objectives proposed in previous studies [29,65], as digital tools create richer learning opportunities for teachers and students, thereby enhancing student engagement and establishing more interaction between teachers and students in the context of digital platforms [30,66,67].

The possible reason for the fluctuation in the interpretation of data skills for SDLC is that such a requirement may be too high for students in vocational institutions. As stated in the previous definition, data skills include the use of learning management systems, digital collaboration tools, and specialized software that support online and blended learning environments. Teachers and students must not only be proficient in using digital tools to improve student engagement and learning outcomes [26]. This may be too high an expectation for these academically disadvantaged groups [42].

One possible explanation for the relative stability of digital literacy in SDLC measurements is that the measurement process covers a range of skills, knowledge, and attitudes, which may be less stringent than the strict requirements for digital skills. This is also consistent with the fact that digital literacy is dynamic and does not fluctuate with factors such as ethnicity and gender. This development is conducive to promoting fairness, inclusiveness, and lifelong learning [45,64].

The two dimensions of digital interaction and technology integration also showed notable results, and one possible explanation is that the use of digital tools to facilitate interactive learning environments and thus increase student engagement was generally recognized by students, which is consistent with previous research [30,66–68].

A more unfortunate point is that the data does not support H4. Students' practical performance is not strongly linked to AI literacy (i.e., the ability to learn digital technologies in a sustainable way). This may be related to the following two reasons. Firstly, it is not clearly documented in the existing literature that one's practical competence on a given problem can be significantly guided by the use of AI or the use of data resources, e.g., as mentioned earlier, AI can be observed in the practice of engineering students enabling them to predict, monitor, optimize, and plan the construction design of a civil engineering project [80], but it does not mean that the that the competencies acquired by the students can be well measured [92]. Secondly, it was also learned through informal interviews with the students that although the students were more enthusiastic about searching for information packages on the Internet, there was a gap between the online information and the practice, and the lack of practical content and videos did not allow the students to understand the practical content very well. This view is supported by some local school teachers, who say that the current rise of AI does not have the ability to integrate digital resources, making it difficult for students to know exactly which resources are key and important. Again, the disadvantage of vocational college students, some students have a reverse psychology to digital learning, as Jiang [41] said, Chinese parents are not willing to let students go to technical schools, resulting in the group of technical school students become a disadvantaged group in China's competitive system, the students will be passive and unable to transform the knowledge they have learned into external capabilities. It is worth noting that practice class grades show a negative effect with having a higher SDLC, and while this pathway is not significant, some effects do appear. One possible explanation is that students are tired of such a model of developing digital learning literacy and the negative effect occurs because of internal resistance. This resistance does not stem from a lack of resources or learning opportunities, but purely from the students' own internal motivation. This explanation can be corroborated with some of Chen's [42] claims that neither parent expects anything from these children because they are not happy to do anything.

   

They lack goals and intrinsic motivation. Another reasonable explanation requires further investigation by researchers. In previous studies [93,94], researchers have raised concerns about whether students will develop a negative reaction to the digital age. Furthermore, teachers' digital literacy has a strong correlation with the development of digital literacy, meaning that students' digital learning abilities and behaviors are constrained by teachers' digital literacy [94].

## Control variables for model impact

This study systematically examined three key control variables: gender, parents' highest level of education, and ethnic attributes (with the core dimension being the comparison between ethnic minorities and the Han Chinese).

In the effect testing of the gender variable, the research model demonstrated a high degree of stability. This stability was not only evident in the unconstrained model (i.e., the baseline model without restrictions on gender-related paths) but was also significant in the expanded model with weight adjustments, indicating that the overall model fit was excellent. Statistical results indicate that the gender factor does not exhibit a statistically significant moderating effect in the model. This finding has important practical implications: it suggests that the predictive effects of artificial intelligence literacy and sustainable digital learning capabilities on student learning behavior are not influenced by gender differences among learners. This conclusion aligns with the findings of previous studies [45,64], further validating the positive role of artificial intelligence technology and digital educational resources in promoting gender equality in education, which aligns closely with the core spirit of the United Nations Sustainable Development Goal 5 (Gender Equality) [69,95]. Additionally, it provides important insights for practical innovation in vocational and adult education, namely, by establishing AI application scenarios and digital resource allocation systems free from gender discrimination, to promote equal opportunities for different gender groups in lifelong learning [95,96].

Notably, the model's robustness was also validated across dimensions of parental educational attainment and ethnic attributes, demonstrating remarkable inclusive characteristics. Specifically, regardless of the highest educational attainment of learners' parents, differences in family economic status, or whether they belong to ethnic minority or Han Chinese groups, all learners can access more equitable and equal high-quality educational resources through the empowerment of artificial intelligence technology and digital resources [69]. This finding breaks through the constraints of family cultural capital and social structural factors on educational opportunities in traditional education, highlighting the unique value of digital technology in mitigating educational inequality.

From the perspective of the dynamic development of educational equity, the widespread application of AI technology can further enhance the breadth and depth of educational participation by establishing personalized learning support systems to ensure that every learner has access to learning opportunities tailored to their needs. Furthermore, when students can effectively use AI tools for self-directed learning, their digital learning capabilities will be systematically enhanced, and this enhancement exhibits balanced distribution across different gender and background groups. Over time, this will help gradually bridge the educational equity gap formed by structural factors such as gender, race, and family economic status in traditional educational models, driving the educational system toward greater inclusivity and balance [45,46].

## Limitations and future work

The findings of this study indicate a stable association between artificial intelligence literacy and sustainable digital learning competencies in terms of learning behavior and actual academic performance, providing empirical support for the theory of the synergistic development of technological literacy and learning competencies. The study further reveals that the influence of artificial intelligence literacy on digital learning behavior exhibits cross-group consistency, unaffected by factors such as gender, family background, or ethnicity, offering a theoretical perspective for understanding new forms of educational equity in the digital age. The inclusion of students from higher vocational colleges in the study population lays a theoretical foundation for constructing a more precise digital literacy assessment system. However, this study has two

limitations: first, the model includes few latent variables, making it difficult to establish effective associations between AI literacy and sustainable digital learning ability at the observed variable level; Second, when measuring sustainable digital learning abilities, relying solely on four artificial intelligence application courses makes it difficult to encourage students to transform sustainable learning abilities into learning behaviors and further internalize them as practical skills. Additionally, since similar artificial intelligence literacy courses have not yet been widely implemented among vocational college students, it is currently impossible to conduct large-scale teaching interventions and effect measurements. Although this study found that students cannot directly influence learning behavior through artificial intelligence literacy and sustainable digital learning competencies, this conclusion still provides several reference directions for future teaching practices. Specifically, vocational colleges can adopt a tiered course design: the foundational tier focuses on strengthening digital tool operation skills (such as the application of learning management systems), the advanced tier emphasizes the cultivation of data integration and analysis abilities, and develop practice-oriented online video resources to bridge the gap between theory and practice. Additionally, it is essential to prioritize students' intrinsic motivation for learning, using project-based learning and AI-related project presentations to stimulate their proactive engagement in learning, thereby reducing the barriers posed by "reverse psychology" to the development of digital learning competencies. As AI technology continues to proliferate, future efforts could include expanding AI-related course offerings and implementing targeted instructional interventions to gradually address the shortcomings of the current curriculum framework.

## Conclusions

This study examined the relationship between AI literacy, sustainable digital learning competencies, students' digital learning behaviors and practical performance with 1004 senior students in Liaoning Province. The results of the study confirm that AI literacy can enhance sustainable digital learning ability, and AI literacy positively affects students' digital learning behaviors through sustainable digital learning ability, but it does not have a strong effect on practice grades. This somewhat supports some previous studies [6–10]. In addition, this paper finds that the use of AI literacy and digital resources can contribute more to the equity of education, which includes not only gender, but also family socioeconomic status, ethnic minorities or Han Chinese, which fits well with the United Nations SDG 4, SDG5.

## Supporting information

**S1 File. software.**
(RAR)

## Author contributions

**Conceptualization:** Xuefei Lin, Guangyu Xu, Bin Xiong.

**Data curation:** Xuefei Lin, Guangyu Xu.

**Formal analysis:** Xuefei Lin, Guangyu Xu.

**Funding acquisition:** Bin Xiong.

**Investigation:** Xuefei Lin.

**Methodology:** Xuefei Lin.

**Project administration:** Xuefei Lin.

**Resources:** Xuefei Lin, Guangyu Xu.

**Software:** Xuefei Lin.

**Supervision:** Xuefei Lin, Guangyu Xu, Bin Xiong.

**Validation:** Guangyu Xu, Bin Xiong.

**Visualization:** Xuefei Lin, Bin Xiong.

**Writing – original draft:** Xuefei Lin, Bin Xiong.

**Writing – review & editing:** Xuefei Lin, Bin Xiong.

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
