## [Decision Letter · Decision Letter 0]

25 Apr 2025

PONE-D-25-07315Artificial Intelligence Literacy, Sustainability of Digital Learning and Practice Achievement: A Structural Equation ModelPLOS ONE

Dear Dr. Xu,

Thank you for submitting your manuscript to PLOS ONE. After careful consideration, we feel that it has merit but does not fully meet PLOS ONE’s publication criteria as it currently stands. Therefore, we invite you to submit a revised version of the manuscript that addresses the points raised during the review process.

The reviewers noted that the title currently emphasises the methodology rather than the core focus of the study and should be revised to reflect the main themes. The introduction and problem statement lack contextual clarity, with references to K–12 education that do not align with the study’s scope. Key constructs such as “sustainable digital learning competencies,” “learning behaviour,” and “educational equity” are insufficiently defined and conceptually underdeveloped. Research questions are double-barrelled and require refinement.

The literature review is disorganised and includes redundancies. The adoption of the COM-B model is not adequately explained or integrated. The conceptual framework lacks clarity and theoretical grounding. Methodological details, including sampling rationale and questionnaire development, are insufficient. The analysis lacks clear links to the research questions, and the use of control variables is not well justified.

The discussion does not follow from the results, and conclusions introduce new terms without prior definition. Implications and limitations are missing. Thank you.

Musa Adekunle Ayanwale

Academic Editor

PLOS ONE

We look forward to receiving your revised manuscript.

Kind regards,

Musa Adekunle Ayanwale

Academic Editor

PLOS ONE

Journal Requirements:

3. Thank you for stating the following financial disclosure: [This research was funded by a grant from the Shanghai Key Laboratory of Pure Mathematics and Mathematical Practice, grant number 18DZ2271000 By XiongBin].

5. Please ensure that you include a title page within your main document. You should list all authors and all affiliations as per our author instructions and clearly indicate the corresponding author.

Reviewers' comments:

Reviewer's Responses to Questions

**Comments to the Author**

1. Is the manuscript technically sound, and do the data support the conclusions?

Reviewer #1: Yes

Reviewer #2: Partly

2. Has the statistical analysis been performed appropriately and rigorously? 

Reviewer #1: Yes

Reviewer #2: Yes

3. Have the authors made all data underlying the findings in their manuscript fully available?

Reviewer #1: Yes

Reviewer #2: No

4. Is the manuscript presented in an intelligible fashion and written in standard English?

Reviewer #1: Yes

Reviewer #2: No

5. Review Comments to the Author

Reviewer #1: Your study effectively highlights the impact of AI on education, particularly in vocational higher education, but some areas could be clearer. The introduction is broad, and specifying what kind of attention vocational education lacks—whether in research, policy, or institutional support—would strengthen your argument. The mention of the COM-B model is useful, but briefly explaining its focus on capability, opportunity, and motivation would help unfamiliar readers. Your methodology is clear, but specifying whether the institution studied is a vocational college would enhance precision. Some findings could be more clearly articulated, such as the statement that AI literacy correlates with sustainable digital learning behaviors through the sustainability of digital learning competences, which could be simplified for better readability. Additionally, the claim that students in vocational institutions struggle to translate learning behavior into good performance in practice needs further clarification—does this mean they lack application skills, institutional support, or workplace opportunities? The idea that sustainability of digital learning competences can be “slightly counterproductive” is intriguing but needs more explanation; in what way does it hinder performance? The connection to SDG5 is interesting, but it is not entirely clear how controlling for gender, family resources, and ethnicity contributes to educational equity—elaborating on this link would strengthen the conclusion. A clearer and more precise articulation of these key points would enhance the impact of your study.

Reviewer #2: I wish to appreciate the effort and thoughtfulness of the authors to present this idea in this changing world. Below are my contributions and concerns:

1. Topic: Artificial Intelligence Literacy, Sustainability of Digital Learning and Practice Achievement: A Structural Equation Model. There is need to align the topic with the topic. The use of analytical tool on the topic is not necessary but making the topic to cover contents and context will be of a great value and attractive to the readers.

2. Introduction: “To enable better use of AI, data science education and data literacy may need to be integrated into K-12 education. Considering the context of higher education and TVET in the opening statements, what is the rationale for introducing the use of AI, data science education, and data literacy into K-12? As Chiu [12] suggests, the relationship between the two is currently not well-tested. What the two in this context?

3. Problem Statement: Problem statement is vague. There is need for critical analysis of literature to identify the gaps for the study.

5. Research Questions: RQ1: Is there a correlation between AI literacy and sustainable digital learning competences? This is good but no background speaks to the dependent variable (sustainable digital learning competence) Please include this.

RQ2: To what extent does AI literacy and the sustainable digital learning competences affect students' learning behaviors? Does this behavior facilitate or help students translate it into practical skills? At beautiful as these questions are, they are double barrel, please split them or transform them to a hypothesis to test the role of AI literacy and sustainable digital learning competences on students’ acquisition of practical skills. Also, provide background to this.

RQ3: Can AI literacy and sustainable digital learning competition contribute more to educational equity (SDG5) in a way? This is good. What is educational equity in the context of this study?

5. Literature: The concepts explained are good, however, most information provided could have been used at the study background.

“digital competence (DC) is interpreted in various ways (such as digital literacy, digital competence, electronic literacy, electronic skills, electronic competence, computer literacy, and media literacy)” This is not clear. There is repetitions here digital competence-digital competence, electronic competence; i think the focus should be what these digital competences are.

We adopt the above concepts and qualify them with the expectation that students will be able to use digital resources sustainably for a long period of time or with a high frequency while learning each skill. This is not clear, what concepts are above. What is learning competences, digital competence and sustainable digital learning competences? How are these measured in literature and specifically, in this study?

No concept clarifications for these: educational equity,

students' learning behaviors, and practical skills.

6.Theoretical Framework: The adoption of COM-B, but the proponent, year, justification, and application of it to current study is vague.

7. Hypothesis: It should be in plural form “Hypotheses” since they are more than one. Also, some of the hypotheses raised are not aligned with research questioned raised at the background. Also, empirical literature was not advanced for based on hypotheses formulated.

8. Framework: There is a spelling error ‘Farmework’ Also, is this conceptual framework or theoretical framework. The figure projects conceptual framework nor theoretical framework. There is need to elaborate how the framework summarises the study and supported by the theory adopted.

9: Ethical Consideration: Move it under the Methodology

10. Methodology: The explanations and presentation are good. However, technically and logically, there are areas for further explanations. What research philosophical and design adopted for the study and rationale for the adoption? What is the population of the study area and why is the adoption of convenience sampling approach? How is the sampling technique adopted provide protect the data and findings from bias? On the questionnaire, was the questionnaire adopted, adapted or researchers developed?

11. Questionnaire: “Investigate students' AI literacy and digital learning skills, and a five-point Likert scale” Where is sustainable digital learning competences, learning behavior, and Practice Achievement? How the data was collected (Online or paper) is missed. The construct “Practice Achievement” What the research has to say here is confusing, which tending towards quasi-experimental approach.

12. Analysis: What is the correlation between “Fig3. SEM of AI Literacy, Sustainable Digital Learning Competencies and Behavior” and “In Model 3, AI literacy and sustainable digital learning competencies and practice achievement”

The study background and literature review did not provide basis for this model. Now, the model tends towards negative, why must the model be retained and explained? The figure for control variables is invisible in the report. No hypotheses or research questions that support the control variables?

13. Discussion: The discussion was not presented logically based on the research hypotheses formulated. Furthermore, the findings were not discussed to establish the relationship that exist between the variables and the implications.

14. Conclusions, Implications and Limitations: The conclusion provided is fair. “This study examined the relationship between AI literacy, sustainable digital learning competencies, students' digital learning behaviors and practical performance with 1004 senior students in Liaoning Province.” However, the introduction of concepts such as students’ digital learning behaviors and practical performance are posing confusion to the readers. The implications of study findings and limitation are missing which are very necessary for study of this type.

6. PLOS authors have the option to publish the peer review history of their article (what does this mean? ). If published, this will include your full peer review and any attached files.

**Do you want your identity to be public for this peer review?** For information about this choice, including consent withdrawal, please see our Privacy Policy .

Reviewer #1: **Yes: ** Dr. Olajumoke Olayemi Salami

Reviewer #2: No

---

## [Author Response · Author response to Decision Letter 1]

30 Jun 2025

Response to Reviewer Comments

Dear Editor and Reviewers,

We sincerely thank you for your thoughtful and constructive comments on our manuscript entitled “Artificial Intelligence Literacy, Sustainability of Digital Learning and Practice Achievement: A Study of Vocational College Students”. We appreciate the time and effort you have invested in reviewing our work.

We have carefully considered all the comments and suggestions provided by the reviewers and have revised the manuscript accordingly. Below, we provide a detailed, point-by-point response to each comment.

Firstly, we have categorized and summarized the comments from Reviewer 1 and identified seven points for revision.

(R11)Your study effectively highlights the impact of AI on education, particularly in vocational higher education, but some areas could be clearer. The introduction is broad, and specifying what kind of attention vocational education lacks—whether in research, policy, or institutional support—would strengthen your argument.

(R12)The mention of the COM-B model is useful, but briefly explaining its focus on capability, opportunity, and motivation would help unfamiliar readers.

(R13)Your methodology is clear, but specifying whether the institution studied is a vocational college would enhance precision.

(R14)Some findings could be more clearly articulated, such as the statement that AI literacy correlates with sustainable digital learning behaviors through the sustainability of digital learning competences, which could be simplified for better readability.

(R15)Additionally, the claim that students in vocational institutions struggle to translate learning behavior into good performance in practice needs further clarification—does this mean they lack application skills, institutional support, or workplace opportunities?

(R16)The idea that sustainability of digital learning competences can be “slightly counterproductive” is intriguing but needs more explanation; in what way does it hinder performance?

(R17)The connection to SDG5 is interesting, but it is not entirely clear how controlling for gender, family resources, and ethnicity contributes to educational equity—elaborating on this link would strengthen the conclusion. A clearer and more precise articulation of these key points would enhance the impact of your study.

Similarly, Reviewer 2's comments are listed below in a point-by-point format.

(R21)Topic: Artificial Intelligence Literacy, Sustainability of Digital Learning and Practice Achievement: A Structural Equation Model. There is need to align the topic with the topic. The use of analytical tool on the topic is not necessary but making the topic to cover contents and context will be of a great value and attractive to the readers.

(R22)Introduction: “To enable better use of AI, data science education and data literacy may need to be integrated into K-12 education. Considering the context of higher education and TVET in the opening statements, what is the rationale for introducing the use of AI, data science education, and data literacy into K-12? As Chiu [12] suggests, the relationship between the two is currently not well-tested. What the two in this context?

(R23) Problem Statement: Problem statement is vague. There is need for critical analysis of literature to identify the gaps for the study.

(R24)Research Questions: RQ1: Is there a correlation between AI literacy and sustainable digital learning competences? This is good but no background speaks to the dependent variable (sustainable digital learning competence) Please include this.

RQ2: To what extent does AI literacy and the sustainable digital learning competences affect students' learning behaviors? Does this behavior facilitate or help students translate it into practical skills? At beautiful as these questions are, they are double barrel, please split them or transform them to a hypothesis to test the role of AI literacy and sustainable digital learning competences on students’ acquisition of practical skills. Also, provide background to this.

RQ3: Can AI literacy and sustainable digital learning competition contribute more to educational equity (SDG5) in a way? This is good. What is educational equity in the context of this study?

(R25)Literature: The concepts explained are good, however, most information provided could have been used at the study background.

“digital competence (DC) is interpreted in various ways (such as digital literacy, digital competence, electronic literacy, electronic skills, electronic competence, computer literacy, and media literacy)” This is not clear. There is repetitions here digital competence-digital competence, electronic competence; i think the focus should be what these digital competences are.

We adopt the above concepts and qualify them with the expectation that students will be able to use digital resources sustainably for a long period of time or with a high frequency while learning each skill. This is not clear, what concepts are above. What is learning competences, digital competence and sustainable digital learning competences? How are these measured in literature and specifically, in this study?

No concept clarifications for these: educational equity,

students' learning behaviors, and practical skills.

(R26)Theoretical Framework: The adoption of COM-B, but the proponent, year, justification, and application of it to current study is vague.

(R27)Hypothesis: It should be in plural form “Hypotheses” since they are more than one. Also, some of the hypotheses raised are not aligned with research questioned raised at the background. Also, empirical literature was not advanced for based on hypotheses formulated.

(R28)Framework: There is a spelling error ‘Farmework’ Also, is this conceptual framework or theoretical framework. The figure projects conceptual framework nor theoretical framework. There is need to elaborate how the framework summarises the study and supported by the theory adopted.

(R29)Ethical Consideration: Move it under the Methodology

(R210) Methodology: The explanations and presentation are good. However, technically and logically, there are areas for further explanations. What research philosophical and design adopted for the study and rationale for the adoption? What is the population of the study area and why is the adoption of convenience sampling approach? How is the sampling technique adopted provide protect the data and findings from bias? On the questionnaire, was the questionnaire adopted, adapted or researchers developed?

(R211)Questionnaire: “Investigate students' AI literacy and digital learning skills, and a five-point Likert scale” Where is sustainable digital learning competences, learning behavior, and Practice Achievement? How the data was collected (Online or paper) is missed. The construct “Practice Achievement” What the research has to say here is confusing, which tending towards quasi-experimental approach.

(R212)Analysis: What is the correlation between “Fig3. SEM of AI Literacy, Sustainable Digital Learning Competencies and Behavior” and “In Model 3, AI literacy and sustainable digital learning competencies and practice achievement”

The study background and literature review did not provide basis for this model. Now, the model tends towards negative, why must the model be retained and explained? The figure for control variables is invisible in the report. No hypotheses or research questions that support the control variables?

(R213)Discussion: The discussion was not presented logically based on the research hypotheses formulated. Furthermore, the findings were not discussed to establish the relationship that exist between the variables and the implications.

(R214)Conclusions, Implications and Limitations: The conclusion provided is fair. “This study examined the relationship between AI literacy, sustainable digital learning competencies, students' digital learning behaviors and practical performance with 1004 senior students in Liaoning Province.” However, the introduction of concepts such as students’ digital learning behaviors and practical performance are posing confusion to the readers. The implications of study findings and limitation are missing which are very necessary for study of this type.

Similarly, the annotations provided by the editor within the manuscript are listed below.

Commented [OS1]: The first sentence is a bit broad. You might clarify how AI's rise is specifically affecting education rather than stating it as a general trend.

Commented[OS2]: Your discussion effectively highlights the challenges and opportunities AI presents in education, particularly in the context of SDG 4. However, some areas could be more concise and precise. The link between AI and inequities in education is well stated, but phrases like “disable some of the ability of students to learn” could be clearer—does this refer to over-reliance on AI or gaps in digital literacy? Additionally, the mention of ethics and inequity could be further connected to specific risks, such as algorithmic bias or accessibility issues. The argument about the lack of rigorous rubrics to assess AI’s impact is strong, but specifying what metrics or frameworks might be needed would add depth. The transition to China’s higher vocational education is relevant, but stating why this group is “not in the mainstream” would

provide clarity. Overall, refining phrasing and strengthening logical connections between AI literacy, sustainable learning, and educational equity would improve the clarity and impact of the section

Commented [OS3]: Your discussion provides a thorough overview of AI literacy, but some areas could be more concise and precise. The definition section is well-supported by literature, but phrases like “educating learners to acquire fundamental concepts, skills, knowledge, and attitudes that do not require a priori knowledge” could be clearer—does this mean AI literacy is accessible without prior expertise? The transition from K-12 to higher education is strong, but clarifying how vocational education specifically fits into this continuum would enhance clarity

Commented [OS4]: Improve clarity, reducing redundancy, and reinforcing logical connections would strengthen this section.

Commented [OS5]: Your discussion effectively contrasts different theoretical models for studying AI adoption and sustainable digital learning behaviors, but it could be more concise.

Commented [OS6]: Your discussion effectively traces the evolution from digital literacy to AI literacy, highlighting their interconnectedness. However, the transition between concepts could be more fluid, particularly in linking digital competence and AI’s role in education. The engineering education example is relevant but could be more concisely integrated to strengthen the argument. The necessity of AI literacy in vocational education is well justified, but reducing

redundancy would improve clarity. The hypothesis (H1) is logical and well-grounded in the preceding discussion.

Commented [OS7]: The statement effectively conveys the reliability and validity of the questionnaires, as well as the strong model fit. However, it could briefly mention any key implications or limitations to provide a more comprehensive conclusion.

We would like to once again express our sincere gratitude to the reviewers and editors for their valuable feedback and efforts. We have summarized all the revision suggestions and integrated the common issues. Our responses are presented below. In the tracked version, some newly added references have led to higher-numbered citations appearing earlier in the text; this will be corrected in the final clean version.

Title:

(R21) Doubts about the readability of titles.

To enhance the readability of the title and avoid research tools appearing in the title, change the title to:

AI Literacy and Sustainability of Digital Learning: Can Vocational Students Translate These into Practical Achievements?

Abstract:

Commented [OS1] Doubts about the scope being too broad

The scope of the first sentence is indeed too broad. Please make the following modifications:

The rapid expansion of AI and the massive use of digital learning are creating a huge change in higher education. In contrast to general higher education, which is at the center of change, the changes in vocational higher education do not seem to have received sufficient attention from researchers.

Introduction:

(R11)

We fully agree that clarifying the specific aspects in which vocational education has been under-addressed would improve the precision and argumentative strength of the introduction. In response, we have revised the introduction to explicitly identify three key dimensions where vocational education remains insufficiently supported:

1.Research dimension: Most existing studies on AI literacy and digital learning focus on general education or higher academic institutions. Vocational education, especially concerning marginalized student groups, remains underrepresented in empirical research.

2.Policy resource allocation: Compared to comprehensive universities, vocational institutions often have limited access to digital infrastructure, educational resources, and faculty development opportunities related to AI and digital learning.

3.Institutional support: There is a noticeable absence of systematic integration of AI tools and data literacy into the curriculum in vocational education, which constrains students' ability to develop sustainable digital learning competencies.

These revisions have been incorporated into the introduction, and we have also improved the connection between this background and our stated research questions. We sincerely appreciate your insightful suggestion, which helped us improve the clarity and relevance of our research focus.

(R22)

We fully understand your concern about the logical coherence and the scope of the study. Upon careful consideration, we agree that this part is not closely related to the main focus of our research, which lies in the context of higher education and vocational education. To maintain the clarity and coherence of the manuscript, we have decided to remove this section entirely. Additionally, the ambiguity related to the citation of Chiu [12] has also been resolved as a result of this deletion. We sincerely appreciate your thoughtful feedback, which has helped improve the focus and rigor of our work.

(R23) (R24) Argument and explanation of research questions

We sincerely thank the reviewer for this insightful comment. In the revised manuscript, we have significantly strengthened the problem statement by integrating a more comprehensive and critical review of existing literature across several educational contexts—K-12, higher education, and vocational education. Specifically: Literature Mapping and Gap Identification: We systematically traced the evolution of AI literacy from digital literacy and identified its multidimensional nature as conceptualized by UNESCO and Ng. While prior studies have investigated AI literacy in K-12 and university settings (Ng [13], Druga et al. [16], Long et al. [22]), we highlighted the limited attention to vocational education, particularly in under-resourced environments. This lack of empirical focus on vocational students represents a concrete research gap. Theoretical and Practical Significance: Building on the existing AI literacy frameworks, we introduced the emerging construct of sustainable digital learning competence and supported it with references from ICILS, OECD, and recent empirical models (Ličen [43]; Yang [41]). We demonstrated that while digital competence and learning competence are increasingly essential in educational success, there is inadequate empirical work connecting these constructs to behavioral outcomes in vocational settings. Contextual Relevance and Marginalization: We expanded the discussion of educational inequality, emphasizing how vocational students in China are structurally marginalized in terms of institutional resources, societal perceptions, and curriculum design. These factors amplify the significance of our study within the Sustainable Development Goals (SDGs) framework, particularly regarding SDG 4, SDG 5, SDG 9, and SDG 10. Explicit Research Questions:

---

## [Decision Letter · Decision Letter 1]

30 Jul 2025

PONE-D-25-07315R1Artificial Intelligence Literacy, Sustainability of Digital Learning and Practice Achievement: A Study of Vocational College StudentsPLOS ONE

Dear Dr. Guangyu Xu,

Thank you for submitting your manuscript to PLOS ONE. After careful consideration, we feel that it has merit but does not fully meet PLOS ONE’s publication criteria as it currently stands. Therefore, we invite you to submit a revised version of the manuscript that addresses the points raised during the review process.

The authors should refine research questions to avoid ambiguity, correct inconsistencies in terminology and framework labelling, clarify research design and measurement details, and expand on theoretical and practical implications. Thank you. Musa Adekunle Ayanwale, Ph.D

Academic Editor

PLOS ONE Please submit your revised manuscript by Sep 13 2025 11:59PM. If you will need more time than this to complete your revisions, please reply to this message or contact the journal office at plosone@plos.org . Please include the following items when submitting your revised manuscript:

We look forward to receiving your revised manuscript.

Kind regards,

Musa Adekunle Ayanwale, Ph.D

Academic Editor

PLOS ONE

Journal Requirements:

Reviewers' comments:

Reviewer's Responses to Questions

**Comments to the Author**

1. If the authors have adequately addressed your comments raised in a previous round of review and you feel that this manuscript is now acceptable for publication, you may indicate that here to bypass the “Comments to the Author” section, enter your conflict of interest statement in the “Confidential to Editor” section, and submit your "Accept" recommendation.

Reviewer #1: All comments have been addressed

Reviewer #2: All comments have been addressed

2. Is the manuscript technically sound, and do the data support the conclusions?

Reviewer #1: Yes

Reviewer #2: Partly

3. Has the statistical analysis been performed appropriately and rigorously? 

Reviewer #1: Yes

Reviewer #2: Yes

4. Have the authors made all data underlying the findings in their manuscript fully available?

Reviewer #1: Yes

Reviewer #2: Yes

5. Is the manuscript presented in an intelligible fashion and written in standard English?

Reviewer #1: Yes

Reviewer #2: Yes

6. Review Comments to the Author

Reviewer #1: (No Response)

Reviewer #2: Dear Author,

I wish to commend your thoughtfulness to conceptualised this timely study especially in the maginalised TVET colleges. The following view comments will enhance the stidy further:

1. RQ2: To what extent does AI literacy and the sustainable digital learning competences affect students' learning behaviors? Does this behavior facilitate or help students translate it into practical skills? This is a double question. What behaviour and how were the practical skills measured in this study?

2. Can AI literacy and sustainable digital learning competition contribute more to educational equity (SDG5) in a way? The rationale for this question is unclear. How it was achieved in the study is not sure. The word "competition" brought a confusion. Hope it is not typographic error.

3. Check this "family socioeconomic status (SES),/ family social status (SES)" in the literature review.

4. COM-B Model- Tag the heading as "theoretical framework" not COM-B Model, beacuse you made used of three models.

5.SDLC - should be fully defined under the hypotheses development before abbreviation.

6. Framework: Is it a conceptual framework/theoretical framework? The title bears conceptual framework while the figure tagged as theoretical framework. It is actually conceptual framework.

7. What design was used? The report of analysis was confusing with the provision made for control variables, the teaching, and assessment by teachers and students among others.

8. The biggest gap in this beautiful study is theoretical and practical implications of findings.

7. PLOS authors have the option to publish the peer review history of their article (what does this mean? ). If published, this will include your full peer review and any attached files.

**Do you want your identity to be public for this peer review?** For information about this choice, including consent withdrawal, please see our Privacy Policy .

Reviewer #1: **Yes: ** Dr Olajumoke Olayemi Salami

Reviewer #2: **Yes: ** Omotayo Adewale AWODIJI

---

## [Author Response · Author response to Decision Letter 2]

19 Aug 2025

Dear Editor and Reviewer,

First of all, our research team would like to express our sincere gratitude to the editors and reviewers for their hard work. We would also like to thank several reviewers for their suggestions for revisions to our research. We have carefully read these suggestions and responded to them in detail. We hope that our responses will adequately address the issues raised in the article.

Below are our responses to the reviewers' comments. We hope that these responses will address your concerns and make our research more coherent and clear.

1. RQ2: To what extent does AI literacy and the sustainable digital learning competences affect students' learning behaviors? Does this behavior facilitate or help students translate it into practical skills? This is a double question. What behaviour and how were the practical skills measured in this study?

Thank you for questioning the research question. Indeed, this statement is not rigorous, so we have made appropriate modifications to it. As we cannot measure to what extent, and we have not set up a control group, we have made the following modifications. And we will address your confusion by carefully and diligently describing the process of measurement and research implementation when answering question 7.

RQ2: Can artificial intelligence literacy and sustainable digital learning competences influence students' learning behavior? Does this behavior facilitate or help students translate it into practical skills?

2. Can AI literacy and sustainable digital learning competition contribute more to educational equity (SDG5) in a way? The rationale for this question is unclear. How it was achieved in the study is not sure. The word "competition" brought a confusion. Hope it is not typographic error.

Thank you for raising your doubts about the research question. Our research team had a strong debate about this issue when we first wrote the paper, because it is difficult to answer and belongs to a problem that cannot provide specific answers and implementation paths. Moreover, there were errors in the writing, and we have made the necessary revisions. The specific modifications are as follows:

RQ3: Can artificial intelligence literacy and sustainable digital learning Competences empower all learners (regardless of family socioeconomic status, gender, ethnicity, etc.) to benefit equally from quality education?

This study systematically examined three key control variables: gender, parents' highest level of education, and ethnic attributes (with the core dimension being the comparison between ethnic minorities and the Han Chinese).

In the effect testing of the gender variable, the research model demonstrated a high degree of stability. This stability was not only evident in the unconstrained model (i.e., the baseline model without restrictions on gender-related paths) but was also significant in the expanded model with weight adjustments, indicating that the overall model fit was excellent. Statistical results indicate that the gender factor does not exhibit a statistically significant moderating effect in the model. This finding has important practical implications: it suggests that the predictive effects of artificial intelligence literacy and sustainable digital learning capabilities on student learning behavior are not influenced by gender differences among learners. This conclusion aligns with the findings of previous studies [45,64] , further validating the positive role of artificial intelligence technology and digital educational resources in promoting gender equality in education, which aligns closely with the core spirit of the United Nations Sustainable Development Goal 5 (Gender Equality) [69,95]. Additionally, it provides important insights for practical innovation in vocational and adult education, namely, by establishing AI application scenarios and digital resource allocation systems free from gender discrimination, to promote equal opportunities for different gender groups in lifelong learning [95,96].

Notably, the model's robustness was also validated across dimensions of parental educational attainment and ethnic attributes, demonstrating remarkable inclusive characteristics. Specifically, regardless of the highest educational attainment of learners' parents, differences in family economic status, or whether they belong to ethnic minority or Han Chinese groups, all learners can access more equitable and equal high-quality educational resources through the empowerment of artificial intelligence technology and digital resources [69]. This finding breaks through the constraints of family cultural capital and social structural factors on educational opportunities in traditional education, highlighting the unique value of digital technology in mitigating educational inequality.

From the perspective of the dynamic development of educational equity, the widespread application of AI technology can further enhance the breadth and depth of educational participation by establishing personalized learning support systems to ensure that every learner has access to learning opportunities tailored to their needs. Furthermore, when students can effectively use AI tools for self-directed learning, their digital learning capabilities will be systematically enhanced, and this enhancement exhibits balanced distribution across different gender and background groups. Over time, this will help gradually bridge the educational equity gap formed by structural factors such as gender, race, and family economic status in traditional educational models, driving the educational system toward greater inclusivity and balance [45,46].

3. Check this "family socioeconomic status (SES),/ family social status (SES)" in the literature review.

Thank you for your suggestion regarding the rigor of terminology. We have revised all references to the term “family socioeconomic status (SES)” to make it more rigorous. After reading several articles, we have decided to use the term “family socioeconomic status.”

Liu, J.; Peng, P.; Luo, L. The Relation between Family Socioeconomic Status and Academic Achievement in China: A Meta-Analysis. Educational Psychology Review 2020, 32, 49–76. https://doi.org/10.1007/s10648-019-09494-0.

Xu, S.; Jin, Y. Chinese Adolescents’ Socioeconomic Status and English Achievement: The Mediating Role of Parental Emotional Support. Humanities and Social Sciences Communications 2024, 11 (1). https://doi.org/10.1057/s41599-024-04048-4.

4. COM-B Model- Tag the heading as "theoretical framework" not COM-B Model, beacuse you made used of three models.

Thank you for your suggestion to change the COM-B model to a theoretical framework. This suggestion is very correct, and we have adopted it. We have indeed used the term “theoretical framework” in the text because this model is more abstract and belongs more appropriately to the category of theoretical frameworks.

The model was proposed by the UK's National Institute for Health and Care Excellence (NIHCE) in 2011, citing the Capabilities, Opportunities, Motivations, Behaviours (COM-B) model [75] as a key theoretical framework for understanding and supporting behaviour change [76].

5.SDLC - should be fully defined under the hypotheses development before abbreviation.

Thank you for your suggestions on the modification of SDLC definition. We have fully considered and made the following modifications:

Based on the above basic concepts and the definitions of digital learning competencies provided by organizations such as UNESCO and the OECD (i.e., a set of knowledge, skills, and attitudes that enable students to effectively use digital tools for learning in digital learning environments), sustainable digital learning competences (SDLC) can be defined as: the ability of individuals to develop habitual skills for long-term or regular use of digital resources and frequent and effective application of digital tools for learning in digital learning environments, based on certain digital competencies and learning abilities.

6. Framework: Is it a conceptual framework/theoretical framework? The title bears conceptual framework while the figure tagged as theoretical framework. It is actually conceptual framework.

Thank you for your feedback on the framework definition. Indeed, COM-B is a theoretical framework. Within this framework, we studied the SDLC and integrated AI literacy with the SDLC to derive a conceptual framework, which represents a specific operational model. The specific modifications are as follows:

Based on the discussion in the previous section, the artificial intelligence literacy questionnaire developed by UNESCO Ieva [8] will be used to measure artificial intelligence literacy. This questionnaire consists of four sections: impact, collaboration, self-reflection, and ethics. The sustainability of digital learning capabilities (SDLC) will be measured using the digital learning capabilities questionnaire developed by Li Cen [26] and Yang [27]. Ieva's [8] and Romero's [88] will be used to determine students' digital learning behaviors, and practice scores will be used to identify the externalized forms of these behaviors. SDLC behaviors are studied based on the COM-B theoretical framework, resulting in the final conceptual framework shown in Figure 1 below.

7. What design was used? The report of analysis was confusing with the provision made for control variables, the teaching, and assessment by teachers and students among others.

Thank you for raising some questions about the research design. Indeed, there are some loopholes in our wording that have caused confusion when reading. Therefore, we have made modifications to make the wording more rigorous and the design process clearer. The specific situation is as follows:

Sustainable digital learning behavior and Practice Achievement

Prior to the distribution of the questionnaire during the spring semester of 2024, students underwent specialized training led by three instructors, focusing on the application of artificial intelligence and digital tools in learning. Concurrently, practical hands-on courses were also implemented in parallel. The training program comprised four courses, with core content covering key modules such as digital resource retrieval and the application of artificial intelligence technology. Students were explicitly required to use artificial intelligence or digital tools for learning activities each week after class, conducting independent searches for relevant materials to support and guide the practical operation courses.

The teaching cycle for courses related to artificial intelligence and digital tools spanned the entire semester. After systematic training, students have developed the ability to independently utilize artificial intelligence and digital resources. Based on the 12-week training principle [25], it can be determined that they have developed sustainable digital learning capabilities.

Following the completion of the aforementioned training and practical courses, the research team organized a final assessment for the practical courses. To ensure the objectivity and professionalism of the assessment results, the test scores were evaluated by three expert instructors with associate senior professional titles, using a percentage-based scoring system (with 60 points as the passing threshold). Since the talent cultivated by this institution primarily targets the heavy industry sector, the relevant practical operations involve numerous high-risk procedures. Improper execution could pose risks to personal safety, leading to stricter evaluation criteria—if experts determine that an operation poses a safety hazard, the test is immediately deemed failed, resulting in a relatively high overall failure rate for the course.

The specific scoring calculation method is shown in Equation (1) below:

(1)

is an independent rating for each teacher. Among them, 323 students failed this test (32.17%), and the remaining 681 passed the test (67.83%), and among the groups that passed the test, the group with a score of 60 to 79 was 327 (32.57%), and the group with a score of 80 to 100 was 354 (35.26%); it is very interesting to note that each of the three groups of failing and good and good each accounted for more than one-third. A reasonable explanation for this is that since there are three teachers grading, if two of them give a low score, they will get a direct fail; if two of them give a pass and one of them gives a fail, they will get a good score; and if all three of them give a pass, they will get a high score.

After the practical assessment was completed, the above questionnaire was distributed, and students self-assessed their sustainable digital learning behavior in the questionnaire. This dimension consisted of three questions adapted from Ieva and Daniela [8] to measure the use of artificial intelligence and digital behavior. Their specific information is as follows:

8. The biggest gap in this beautiful study is theoretical and practical implications of findings.

Thank you for your questions regarding the theoretical and practical significance of the article. After careful consideration by our research team, we have decided to summarize the theoretical and practical significance in the limitations and future work sections and have rewritten this part. The specific details are as follows:

The findings of this study indicate a stable association between artificial intelligence literacy and sustainable digital learning competencies in terms of learning behavior and actual academic performance, providing empirical support for the theory of the synergistic development of technological literacy and learning competencies. The study further reveals that the influence of artificial intelligence literacy on digital learning behavior exhibits cross-group consistency, unaffected by factors such as gender, family background, or ethnicity, offering a theoretical perspective for understanding new forms of educational equity in the digital age. The inclusion of students from higher vocational colleges in the study population lays a theoretical foundation for constructing a more precise digital literacy assessment system. However, this study has two limitations: first, the model includes few latent variables, making it difficult to establish effective associations between AI literacy and sustainable digital learning ability at the observed variable level; Second, when measuring sustainable digital learning abilities, relying solely on four artificial intelligence application courses makes it difficult to encourage students to transform sustainable learning abilities into learning behaviors and further internalize them as practical skills. Additionally, since similar artificial intelligence literacy courses have not yet been widely implemented among vocational college students, it is currently impossible to conduct large-scale teaching interventions and effect measurements. Although this study found that students cannot directly influence learning behavior through artificial intelligence literacy and sustainable digital learning competencies, this conclusion still provides several reference directions for future teaching practices. Specifically, vocational colleges can adopt a tiered course design: the foundational tier focuses on strengthening digital tool operation skills (such as the application of learning management systems), the advanced tier emphasizes the cultivation of data integration and analysis abilities, and develop practice-oriented online video resources to bridge the gap between theory and practice. Additionally, it is essential to prioritize students' intrinsic motivation for learning, using project-based learning and AI-related project presentations to stimulate their proactive engagement in learning, thereby reducing the barriers posed by “reverse psychology” to the development of digital learning competencies. As AI technology continues to proliferate, future efforts could include expanding AI-related course offerings and implementing targeted instructional interventions to gradually address the shortcomings of the current curriculum framework.

---

## [Editor Report · Decision Letter 2]

27 Aug 2025

Artificial Intelligence Literacy, Sustainability of Digital Learning and Practice Achievement: A Study of Vocational College Students

PONE-D-25-07315R2

Dear Dr. Guangyu,

We’re pleased to inform you that your manuscript has been judged scientifically suitable for publication and will be formally accepted for publication once it meets all outstanding technical requirements.

Kind regards,

Musa Adekunle Ayanwale, Ph.D

Academic Editor

PLOS ONE

Additional Editor Comments (optional):

The authors have diligently addressed all the concerns raised by the reviewer, resulting in an improved quality of the work. Thank you.
---

## [Editor Report · Acceptance letter]

PONE-D-25-07315R2

PLOS ONE

Dear Dr. Xu,

I'm pleased to inform you that your manuscript has been deemed suitable for publication in PLOS ONE. Congratulations! Your manuscript is now being handed over to our production team.

Kind regards,

on behalf of

Dr Musa Adekunle Ayanwale

Academic Editor

PLOS ONE